# Placental sex-dependent spermine synthesis regulates trophoblast gene expression through acetyl-coA metabolism and histone acetylation

Irving L. M. H. Aye [1,2✉], Sungsam Gong [1,2], Giulia Avellino [1,2], Roberta Barbagallo[1], Francesca Gaccioli [1,2], Benjamin J. Jenkins [3], Albert Koulman [3], Andrew J. Murray [2,4], D. Stephen Charnock-Jones [1,2,5] & Gordon C. S. Smith [1,2,5]

Placental function and dysfunction differ by sex but the mechanisms are unknown. Here we show that sex differences in polyamine metabolism are associated with escape from X chromosome inactivation of the gene encoding spermine synthase (SMS). Female placental trophoblasts demonstrate biallelic SMS expression, associated with increased SMS mRNA and enzyme activity. Polyamine depletion in primary trophoblasts reduced glycolysis and oxidative phosphorylation resulting in decreased acetyl-coA availability and global histone hypoacetylation in a sex-dependent manner. Chromatin-immunoprecipitation sequencing and RNA-sequencing identifies progesterone biosynthesis as a target of polyamine regulated gene expression, and polyamine depletion reduced progesterone release in male tropho-blasts. The effects of polyamine depletion can be attributed to spermine as SMS-silencing recapitulated the effects on energy metabolism, histone acetylation, and progesterone release. In summary, spermine metabolism alters trophoblast gene expression through acetyl-coA biosynthesis and histone acetylation, and SMS escape from X inactivation explains some features of human placental sex differences.

[1] Department of Obstetrics and Gynaecology, University of Cambridge, NIHR Cambridge Biomedical Research Centre, Cambridge, UK. [2] Centre for Trophoblast Research (CTR), Department of Physiology, Development and Neuroscience, University of Cambridge, Cambridge, UK. [3] NIHR BRC Core Metabolomics and Lipidomics Laboratory, University of Cambridge, Cambridge, UK. [4] Department of Physiology, Development and Neuroscience, University of Cambridge, Cambridge, UK. [5] These authors contributed equally: D. Stephen Charnock-Jones, Gordon C. S. Smith. ✉email: ia319@medschl.cam.ac.uk

Maternal and perinatal deaths account for 6–7% of all deaths globally[1]. Placental dysfunction is implicated in the aetiology of many of the associated conditions, including preeclampsia and foetal growth restriction. Foetal sex influences both the incidence and outcome of pregnancies affected by these disorders. As a foetal organ, the placenta also exhibits sex-differences in its function. Although the underlying causes of placental sex differences are still incompletely understood, sex hormones and genetic factors are likely to play major roles.

The genetic and epigenetic factors that contribute to placental sexual dimorphism largely centre on the X chromosome. The X chromosome can contribute to sex-differences in gene expression by means of escape from X chromosome inactivation (XCI). During XCI, one X chromosome is selected (randomly in humans) for transcriptional silencing. However, as many as 23% of X-linked genes have been reported to escape XCI in humans (depending on tissue type)[2], and to display biallelic expression and increased gene expression in females. We previously hypothesized that XCI escape contributes to the female-bias in placental expression of X-linked genes[3]. However, little is known about the functional consequences of the genes escaping XCI, especially as it relates to placental function and its relationship with pregnancy disorders.

Our previous study showed dysregulated placental polyamine metabolism with pregnancy disorders preeclampsia and foetal growth restriction[3]. Moreover, we reported foetal sex-dependent associations in the polyamine end-metabolite N1,N12-diacetyl-spermine (DAS) in the serum of pregnant women with these placenta-related pregnancy complications[3]. Polyamines are ubiquitous polycations that are essential for cell growth, differentiation, and survival[4]. Polyamines regulate gene expression through wide-ranging mechanisms including direct binding to nucleic acids, translation initiation, and interacting with transcription factors and epigenetic processes[5]. In mammalian cells, polyamines consist of putrescine, spermidine, and spermine. Putrescine is generated from ornithine by ornithine decarboxylase (ODC) in the rate-limiting step for polyamine biosynthesis. Putrescine is converted to spermidine by spermidine synthase (SRM) and subsequently, spermidine is converted to spermine via spermine synthase (SMS). Spermidine and spermine can also be catabolized by the enzyme spermidine/spermine acetyltransferase (SSAT) into N1-acetyl-spermidine (NAS) and DAS respectively, which facilitates their cellular export.

Polyamine biosynthesis is upregulated during placentation[6], and knockout or pharmacological inhibition of ODC in rodents results in embryo lethality[7] and abnormal placental development[8]. Moreover, human infants with loss-of-function mutations in spermine synthase (SMS), a condition known as Snyder Robinson Syndrome (SRS), are born small-for-gestational age (SGA)[9,10]. Despite these profound phenotypes, it is still unclear how impairments in polyamine metabolism cause these effects.

In this study, we tested the hypothesis that SMS escapes XCI resulting in biallelic expression in female placentas and sex-biased differences in polyamine metabolism. We further investigate the functional consequences of this sex difference and show that polyamines regulate mitochondrial metabolism. Moreover, mitochondrial insufficiency induced by polyamine depletion reduced acetyl-CoA availability and histone acetylation, with greater effect in male trophoblasts, resulting in altered expression of genes regulating progesterone synthesis and decreased progesterone secretion.

## Results

### Sex differences in polyamine metabolism are associated with XCI escape of SMS. We previously showed that the female-bias

in placental polyamine metabolism was associated with higher SMS expression in human placental tissues[3]. Moreover, we hypothesized that SMS escapes XCI based on indirect evidence showing female-biased expression and methylation patterns[3]. However, direct evidence of XCI escape requires demonstration of SMS expression from both copies of the X chromosome (i.e., active and inactive X). Therefore, we investigated biallelic expression of SMS at the single cell level in placental tissues to account for random XCI. Because the syncytiotrophoblast layer, which makes up the majority of trophoblasts in the term placenta, is a multinucleated epithelium, isolation of single cells is not possible. Therefore we isolated single nuclei from placental tissues, and took advantage of heterozygous single nucleotide polymorphisms (hetSNPs) present within the mature mRNA of the SMS gene, enabling the identification of the mRNAs encoded by the two copies of X chromosomes in females.

Using our placenta transcriptome data set[11], we identified four female placentas from 90 analyzed, which contained a hetSNP in an exonic region of the SMS gene (position X:21940733A/G). The number of placentas identified for this hetSNP correlates with the overall prevalence of the hetSNP in the European population (minor allele frequency: 0.26)[12] and the low sequencing coverage of the SNP which is located in the 5'UTR region[13]. As an internal control, we identified an exonic hetSNP in TIMM17B (position X:48894188G/A), an X-linked gene that was predicted to be X inactivated (i.e., does not show female-biased expression). The selected female placentas were heterozygous at both positions. Intact nuclei from each placenta were isolated, sorted into a single nucleus per well and the cDNA from each nucleus was obtained as illustrated in Fig. 1a. To detect both alleles in each nucleus, duplex TaqMan SNP genotyping assays were performed whereby each allele-specific primer and probe set are labelled with different fluorescent dyes, one for each allele. To confirm the predicted genotype inferred from the RNA-sequencing (RNA-seq), the corresponding umbilical cord tissue was genotyped (Supplemental Fig. 1a). Following multiplex qPCR analysis using allele-specific primers the individual nuclei were called as expressing the reference, alternate, or both alleles (biallelic). Using our female placentas containing the hetSNP, we observed biallelic expression of SMS in 100% of the placentas (4 out of 4) confirming that SMS was expressed from both alleles in single nuclei. Consistent with previous reports demonstrating a high level of variability in XCI escape between individual cells[2,14–17], 4–40% of the individual nuclei exhibited biallelic expression for this hetSNP (Fig. 1b and Supplemental Fig. 1b). On the other hand, single nuclei expressed either the reference or the alternative TIMM17B alleles but never both in the same nucleus, confirming X inactivation of this gene (Fig. 1b). These findings provide qualitative evidence of biallelic SMS expression and confirm that SMS escapes XCI resulting in biallelic expression in female placentas.

We next explored the functional significance of female biallelic SMS expression by determining the mRNA level and activity of SMS using isolated primary human trophoblasts (PHTs). Compared to male PHTs, female PHTs express 50% higher levels of SMS mRNA (Fig. 1c) and approximately three-fold higher levels of DAS, the polyamine metabolic end-product catalyzed from spermine (Fig. 1d). This demonstrates that the female biallelic SMS expression is associated with increased flux of spermine resulting in its end-product DAS.

**Inhibition of polyamine biosynthesis profoundly alters the PHT transcriptome with enrichment of genes involved in mitochondrial metabolism.** The specific role of polyamines on placental function was investigated by depleting all polyamines

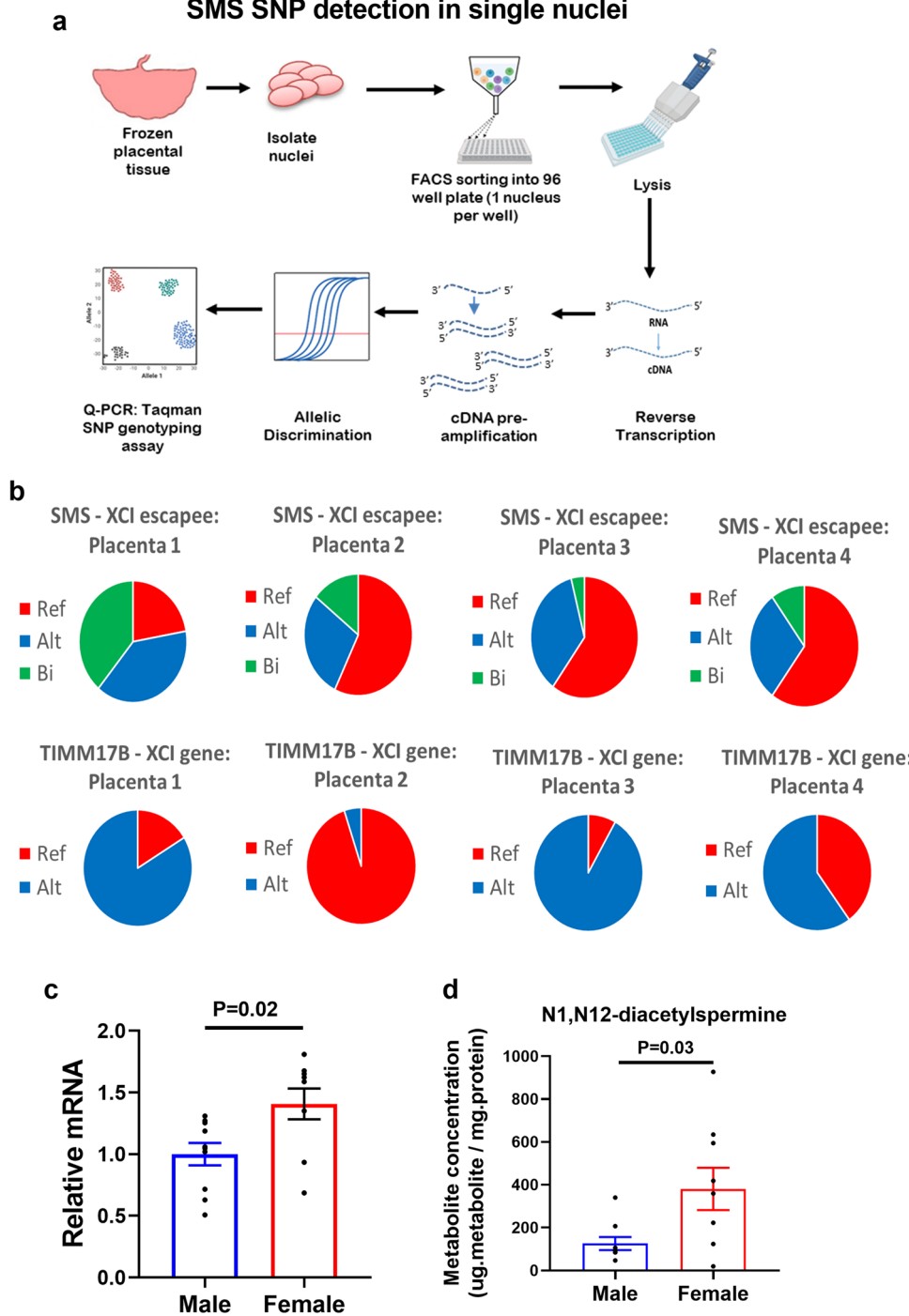

**Fig. 1 Sex differences in polyamine metabolism are associated with XCI escape of SMS. a** Experimental workflow to assess XCI escape of SMS in single nuclei in female placental tissues. Nuclei were isolated from placental tissues and FACS sorted into 96 well plates with one nucleus per well. In the same well, nuclei were lysed, DNAse treated and mRNA reverse transcribed to cDNA, including a preamplification of the regions of interest. The resulting cDNA was then used for SNP typing using multiplexed TaqMan probes specific for both the reference and alternative alleles and allelic discrimination performed in each nuclei by qPCR. **b** Monoallelic and biallelic expression of SMS (XCI escapee) and TIMM17B (XCI inactivated gene) SNPs in single placental nuclei. **c** SMS mRNA levels in PHTs. **d** Polyamine metabolite N1,N12-diacetylspermine concentrations in PHTs. Bar graphs show mean ± SEM, $N = 9$–10 male PHTs and $N = 9$–10 female PHTs. The panel (**a**) was created with Biorender.com.

from PHTs by inhibiting ODC using difluoromethylornithine (DFMO). We previously showed that polyamine depletion with DFMO led to concentration-dependent reductions in cell viability which was more severe in male PHTs[3]. We chose the lowest concentration (5 mM) of DFMO that mediated sex-specific responses for further analyses. DFMO treatment led to

significant reductions in spermidine and spermine, and the reductions in spermine were greater in male PHTs (Supplemental Fig. 2a).

To determine the effects of polyamine depletion on gene expression, we performed RNA-seq in PHTs following DFMO treatment. This analysis revealed profound changes in the PHT

transcriptome of both sexes, with 4710 differentially expressed genes (DEGs) in male PHTs and 3558 DEGs in female PHTs (Supplemental Data 1 and 2), with an overlap of 2689 DEGs between males and female PHTs (overlap $P$-value $= 1.7 \times 10^{-69}$). The RNA-seq results were confirmed by qPCR analysis of a subset of genes in independent biological replicates (Supplemental Fig. 2b and Supplemental Data 5).

To gain insights into the biological functions of the DEGs, gene set enrichment analysis (GSEA[18]) was performed. Polyamine depletion resulted in significant enrichment of 9 pathways in male PHTs and 4 pathways in female PHTs, in the Hallmarks gene sets (Supplemental Fig. 3a); and 5 pathways in male PHTs and 2 pathways in female PHTs, in the KEGG gene sets (Supplemental Fig. 3b). Genes associated with the tricarboxylic acid (TCA) cycle and Oxidative Phosphorylation (OXPHOS) were highly ranked in both the Hallmarks and KEGG gene sets (Supplemental Fig. 3a, b), suggesting alterations in mitochondrial metabolism following polyamine metabolism.

To further investigate the link between polyamines and mitochondrial metabolism, we measured TCA cycle intermediates and polyamine metabolites in placental tissues from healthy pregnancies by liquid chromatography mass spectrometry (LC-MS) and performed correlation analyses. The polyamine catabolic end products DAS (Fig. 2a) and N1-acetylspermidine (NAS) (Fig. 2b) were positively correlated with all 8 TCA cycle intermediates in both male and female placentas. These correlations were striking with values of Pearson's r between 0.57 and 0.77 and adjusted $P$-values between $1.3 \times 10^{-10}$ and $2.5 \times 10^{-21}$.

**Polyamines regulate central energy metabolism.** Given the strong correlations between polyamines and TCA cycle intermediates, we asked whether polyamines regulate the TCA cycle. We measured the expression of TCA cycle enzymes and levels of TCA cycle intermediates in PHTs depleted of polyamines (Fig. 3a). Consistent with our RNA-seq results, DFMO significantly decreased the mRNA levels of 7 out of the 8 TCA cycle enzymes in male PHTs and 5 enzymes in female PHTs (Fig. 3b). Similarly, DFMO treatment also led to significant reductions in TCA cycle intermediates in both male PHTs (6 out of 8 metabolites) and female PHTs (3 out of 8 metabolites) (Fig. 3c).

We then determined whether the effects of polyamines extend towards functional changes in the two major cellular ATP-generating pathways, mitochondrial OXPHOS, and glycolysis. ODC inhibition with DFMO led to significant reductions in OXPHOS and glycolysis in male PHTs but not females (Fig. 4a, b; Supplemental Fig. 4a). The interaction between sex and treatment were statistically significant in the effects of DFMO on OXPHOS but not glycolysis.

To confirm that the effects were due to polyamine depletion and not related to potential off-target effects of DFMO, we used additional approaches to deplete polyamines including SMS silencing (to inhibit spermine synthesis) and SSAT-induction with DENSPM (to activate polyamine catabolism[19]). Supplemental Figure 5a, b shows siRNA knockdown efficiencies and Supplemental Fig. 5c, d demonstrates the effects of SMS-silencing and SSAT-induction on polyamine levels. SMS-silencing reduced OXPHOS and glycolysis in both male and female PHTs (Fig. 4c, d and Supplemental Fig. 4b). SSAT induction with DENSPM decreased OXPHOS and glycolysis in both male and female PHTs (Fig. 4e, f and Supplemental Fig. 4c). DENSPM specifically targeted SSAT as its effects were reversed by silencing SAT1 (gene encoding SSAT; Fig. 4e, f). These results indicate that depletion of polyamines leads to significant suppression of OXPHOS and glycolysis, with a greater effect in male PHTs compared to female PHTs. The sex-specific effects depend on the targeted enzyme

and were not evident when spermine synthesis is specifically downregulated.

**Acetyl-CoA availability links polyamine metabolism to histone acetylation and gene expression.** Previous studies have shown that cellular bioenergetic status is linked to transcriptional remodelling associated with histone acetylation through the availability of acetyl-CoA[20,21]. Histone acetyltransferases (HATs) utilize acetyl-CoA which is their rate-limiting substrate. Thus, cellular acetyl-CoA levels profoundly influence HAT activity, and subsequently, histone acetylation. Given the profound changes in both gene expression and bioenergetics with polyamine depletion, we postulated that suppression of energy metabolism in PHTs treated with DFMO could lead to decreased acetyl-CoA levels and global histone hypoacetylation. Consistent with this hypothesis, DFMO significantly reduced acetyl-CoA levels in male but not female PHTs (Fig. 5a). We then measured the lysine acetylation of several H3 histone marks by immunoblotting analyses following polyamine depletion. DFMO decreased H3K9 and H3K27 acetylation in male PHTs but not females, while H3K14Ac and H3K18Ac were not affected in either sex (Fig. 5b). We focused on H3K27Ac because it is a robust mark of active promoters and distal enhancers that are tightly coupled to gene expression and transcription factor binding[22,23], and previous studies had shown that alterations in this histone mark was associated with foetal growth restriction[24]. Increased spermine catabolism induced by DENSPM reduced H3K27Ac abundance in both male and female PHTs, and this effect was reversed by silencing SAT1 (Fig. 5c). Similarly, H3K27 acetylation was decreased following SMS-silencing in both male and female PHTs (Fig. 5d), suggesting that these effects are mediated by spermine.

Histone acetylation state is dependent on the balance between acetylation by HATs and deacetylation by histone deacetylases (HDACs). Therefore, the possibility exists that polyamine depletion decreased histone acetylation by HDAC activation. To test this alternative hypothesis, we pre-treated PHTs with the global HDAC inhibitor (trichostatin A, TSA), followed by DFMO. We reasoned that if the effects of polyamine depletion by DFMO were mediated by HDACs, then DFMO treatment would not change histone acetylation upon HDACs inhibition (Supplemental Fig. 6a). However, HDAC inhibition by TSA increased H3K27 acetylation that was partially reversed by DFMO treatment (Supplemental Fig. 6b), suggesting that HDACs do not play a clear role in mediating polyamine's effects. This data suggests that the effects of polyamine depletion on histone acetylation are not likely to be mediated by HDAC activity, but rather by regulating acetyl-CoA availability.

**Genome-wide profiling of H3K27Ac identifies polyamine metabolism sensitive genes.** We next assessed which epigenomic loci were most sensitive to polyamine levels by performing H3K27Ac chromatin-immunoprecipitation sequencing (ChIP-seq) analysis of vehicle and DFMO-treated PHTs. We obtained a median of 15 million reads across all samples (ChIPs and Inputs) and found 71,231 consensus peak regions where the majority (>58%) were annotated within the promoter regions, i.e., within 2 kb of TSS (Fig. 6a and Supplementary Fig. 7a). Unsupervised principal component analysis was performed to examine the relatedness in H3K27Ac enriched DNA regions by DFMO treatment and foetal sex. This multivariate analysis shows that H3K27Ac peaks were clearly separated by treatment in male PHTs (i.e., vehicle vs DFMO) but not in females (Fig. 6b), consistent with a sex-biased effect of polyamine depletion.

We identified differentially bound regions (DBRs), separately for male and female PHTs (see "Methods" for details). In male PHTs,

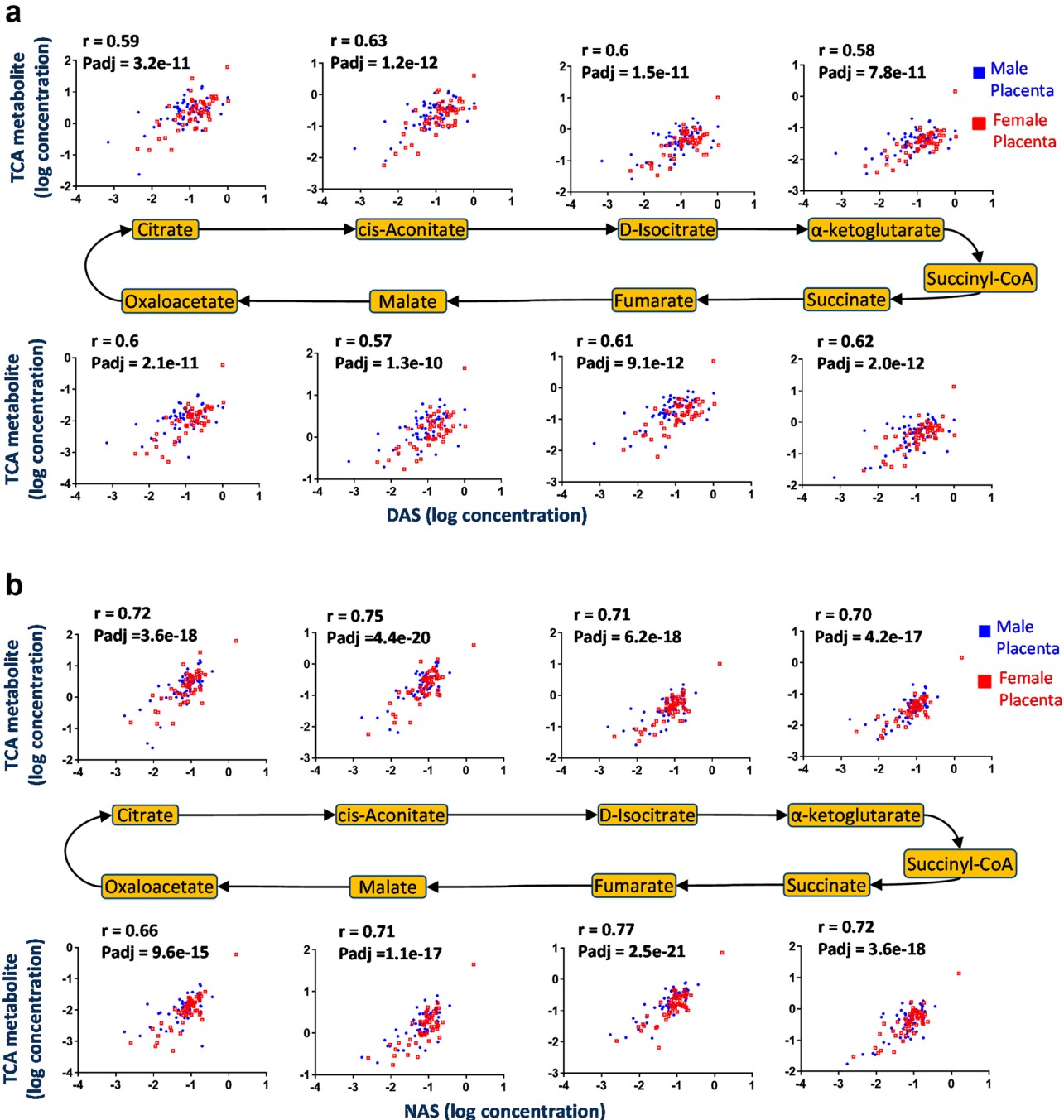

**Fig. 2 Correlations between TCA cycle intermediates and polyamine end metabolites.** Scatter plots of TCA cycle intermediates versus (**a**) DAS or (**b**) NAS in human placental tissues. Metabolite concentrations measured by LC-MS were log transformed and Pearson's correlation analyses performed. $r$ = Pearson's correlation coefficient. $P$-adj = $P$-values corrected for FDR by Benjamini–Hochberg method. DAS: N1,N12-diacetylspermine; NAS: N1-acetylspermidine. $N = 59$ male placentas and $N = 51$ female placentas.

we found 213 and 147 DBRs based on edgeR and DESeq2 respectively, representing 188 and 133 genes (Supplemental Data 3 and 4). No DBRs were identified in female PHTs confirming the sex-biased effect of DFMO. Importantly, 90% of the male DBRs were associated with decreased acetylation, consistent with the global decreases in H3K27Ac levels examined by immunoblotting.

We then determined if these histone acetylation changes were associated with alterations in gene expression. We restricted our analyses to DBRs identified by both edgeR and DESeq2, thus resulting in 146 common DBRs corresponding to 115 genes. By integrating with the RNA-seq data generated in untreated and

DFMO-treated male PHTs, 35 (~30%) of the genes with DBRs had altered mRNA levels (Fig. 6c). After excluding DBRs in intron or intergenic regions, 17 DBR-represented genes remained, of which 16 were downregulated by DFMO. Three of these downregulated genes are involved in the regulation of steroid hormone biosynthesis: INSIG1, which regulates cholesterol biosynthesis thereby providing the backbone for all steroid hormones; HSD3B1, an enzyme that catalyses pregnenolone to progesterone; and SLCO2B1 which transports conjugated steroid hormones (Supplemental Fig. 7b). These DNA regions along with several other DBRs identified in our analysis were validated in independent biological

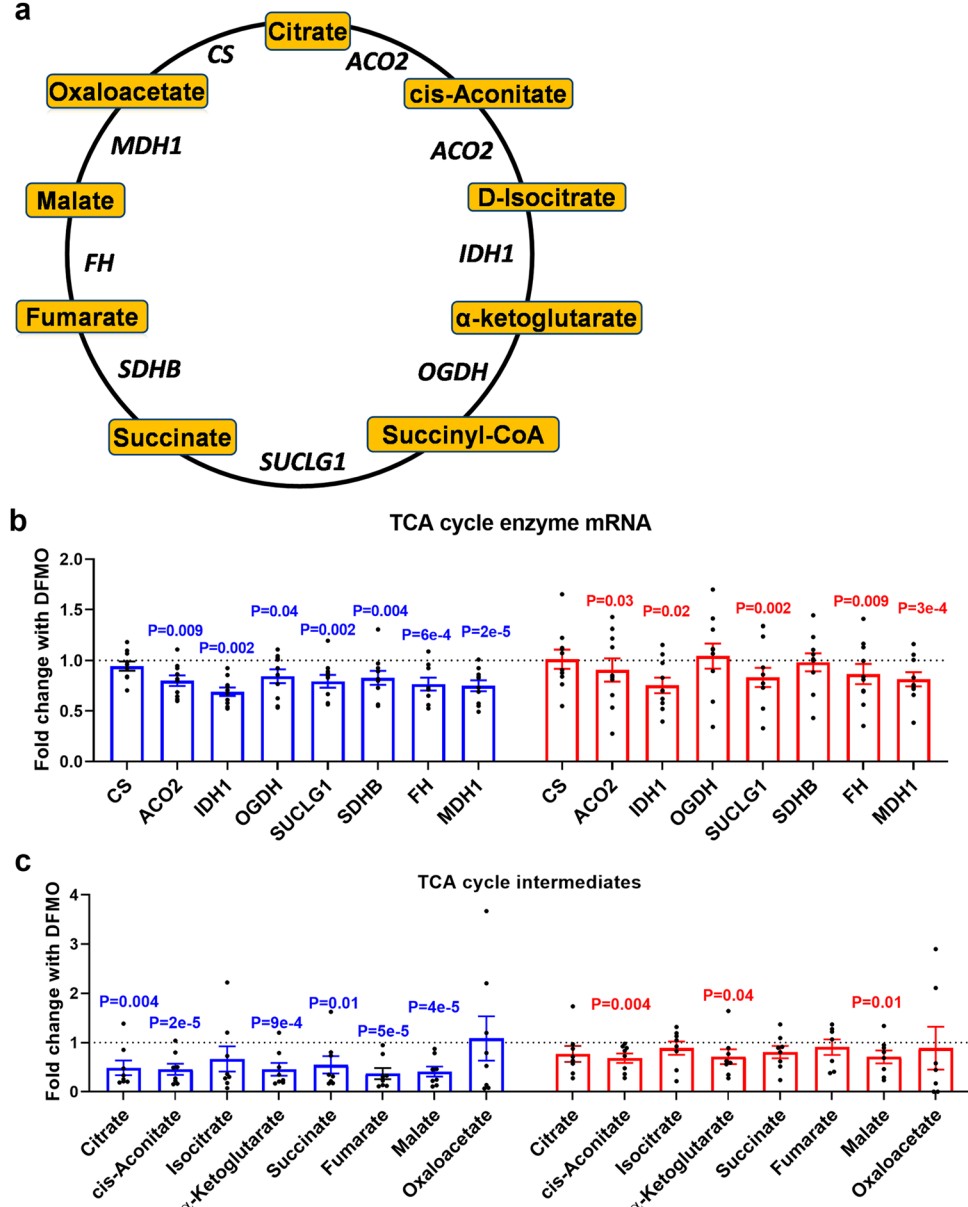

**Fig. 3 Polyamine metabolism is linked to TCA cycle metabolism. a** Illustration of the TCA cycle indicating the metabolites and enzymes measured. **b** TCA cycle enzyme mRNA measured by Q-PCR and (**c**) TCA cycle metabolites measured by LC-MS following DFMO-mediated polyamine depletion in PHTs. Bar graphs show mean ± SEM, $N = 8$–$10$ male PHTs and $N = 8$–$10$ female PHTs.

replicates by ChIP-qPCR assays (Supplemental Fig. 7b). We focused on HSD3B1 as this enzyme plays a major role in placental progesterone synthesis[25]. Consistent with the reduced histone acetylation in the regulatory regions of HSD3B1 (Fig. 6d), both HSD3B1 mRNA (Fig. 6e) and progesterone release (Fig. 6f) were significantly reduced by polyamine depletion by DFMO in male PHTs but not female PHTs. Moreover, the interaction between sex and treatment was significant. SMS-silencing reproduced these effects, resulting in decreased HSD3B1 mRNA and progesterone release (Supplemental Fig. 8a, b). Taken together, these findings suggest that global histone hypoacetylation mediated by depletion of polyamines including spermine leads to physiological changes associated with altered endocrine activity in male PHTs.

## Discussion

In this study, we examined the functional consequences of placental sex differences in polyamine metabolism[3]. We report that

an XCI escapee (SMS) contributes to sex-related differences in placental function. Through a series of biochemical studies, transcriptomic, and epigenomic profiling, our results suggest that the sex-bias in polyamine, and in particular spermine synthesis, influences histone acetylation by regulating the availability of the acetyl donor, acetyl-CoA, through its effects on mitochondrial metabolism (Fig. 7a).

Our previous study predicted that 47 X-linked genes escape inactivation[3], based on female-biased mRNA expression (i.e., higher levels in females compared to males) and DNA methylation (i.e., lack of promoter hypermethylation in females compared to males). These analyses identified SMS as a putative XCI escapee. However, investigating sex-biased expression and DNA methylation are proxies for XCI status and direct measurement of XCI can only be demonstrated at the single cell (or nucleus) level. By investigating XCI escape in single nuclei, we show heterogenous biallelic SMS expression in female placentas that was

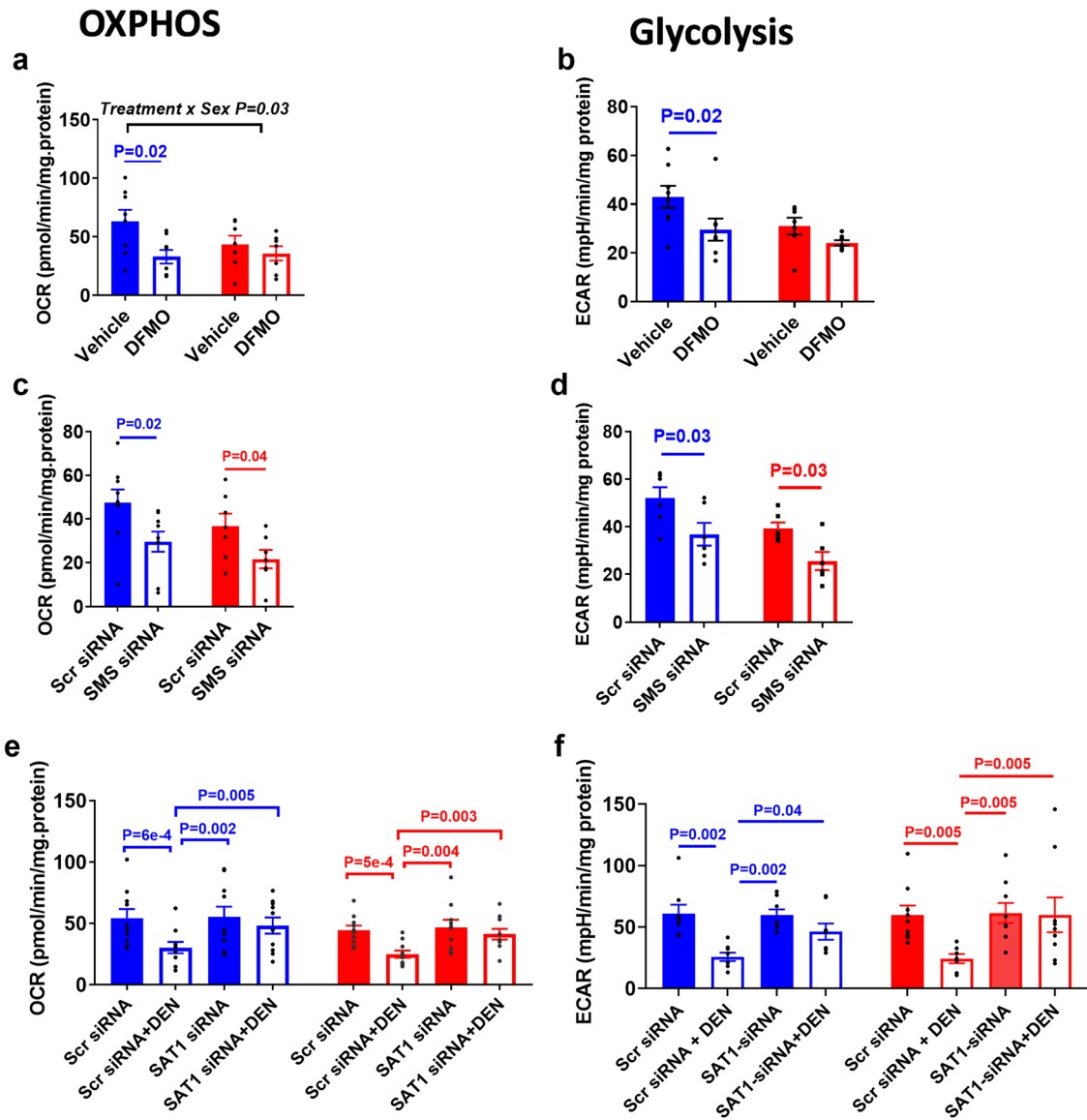

**Fig. 4 Suppression of cellular energy metabolism by polyamine depletion.** Polyamine depletion by DFMO decreases (**a**) OXPHOS and (**b**) glycolysis in male PHTs. Inhibition of spermine synthesis by SMS-silencing decreases (**c**) OXPHOS and (**d**) glycolysis in both male and female PHTs. Induction of SSAT-mediated polyamine catabolism by DENSPM decreases (**e**) OXPHOS and (**f**) glycolysis in both male and female PHTs. Bar graphs show mean ± SEM, $N = 8$–10 male PHTs and $N = 8$–10 female PHTs. OCR oxygen consumption rate; ECAR extracellular acidification rate; were measured on a Seahorse bioanalyzer.

present in up to 40% of cells, which is consistent with previous reports of variable X-inactivation in human placentas[17]. Importantly, the biallelic SMS expression was associated with higher SMS mRNA levels and greater concentrations of DAS, the catabolic end-metabolite of spermine, in female PHTs indicating increased spermine metabolism.

Polyamines are involved in many biological processes that are often varied and wide-ranging, and a complete mechanistic understanding of how polyamines exert their functions is presently lacking. Thus, we sought to clarify the role of polyamines and in particular, the functional significance of sex-differences in placental polyamine metabolism. We targeted several enzymes in the polyamine metabolic pathway using pharmacological and molecular approaches and found that polyamine depletion by DFMO was the only approach that produced sex-specific effects. Although not all effects of DFMO treatment were statistically different when tested for sex by treatment interaction, the effect sizes were larger in males compared to females due to female-biased SMS expression which

compensates for spermine depletion. For example, DFMO decreased intracellular levels of spermidine and spermine (putrescine was not detectable) but the female-biased SMS expression was able to buffer some of the effects of polyamine depletion, resulting in higher spermine concentrations in female PHTs compared to male PHTs following DFMO treatment. Notably, depletion of spermine by SMS-silencing recapitulated many of the effects of polyamine depletion by DFMO but produced similar effects in both male and female PHTs. This can be explained due to the fact that the female bias in SMS expression is the source of the placental sex bias in polyamine metabolism. The polyamine metabolic pathway is highly regulated under physiological conditions, and therefore it is likely that the functional consequences of the sex-bias in spermine synthesis become apparent under polyamine limiting conditions (Fig. 7B). Consistent with this hypothesis, previous studies demonstrate that the sex-specific effects of another placenta-specific XCI escapee, O-linked N-acetylglucosamine transferase (OGT), are unmasked by maternal stress[26,27].

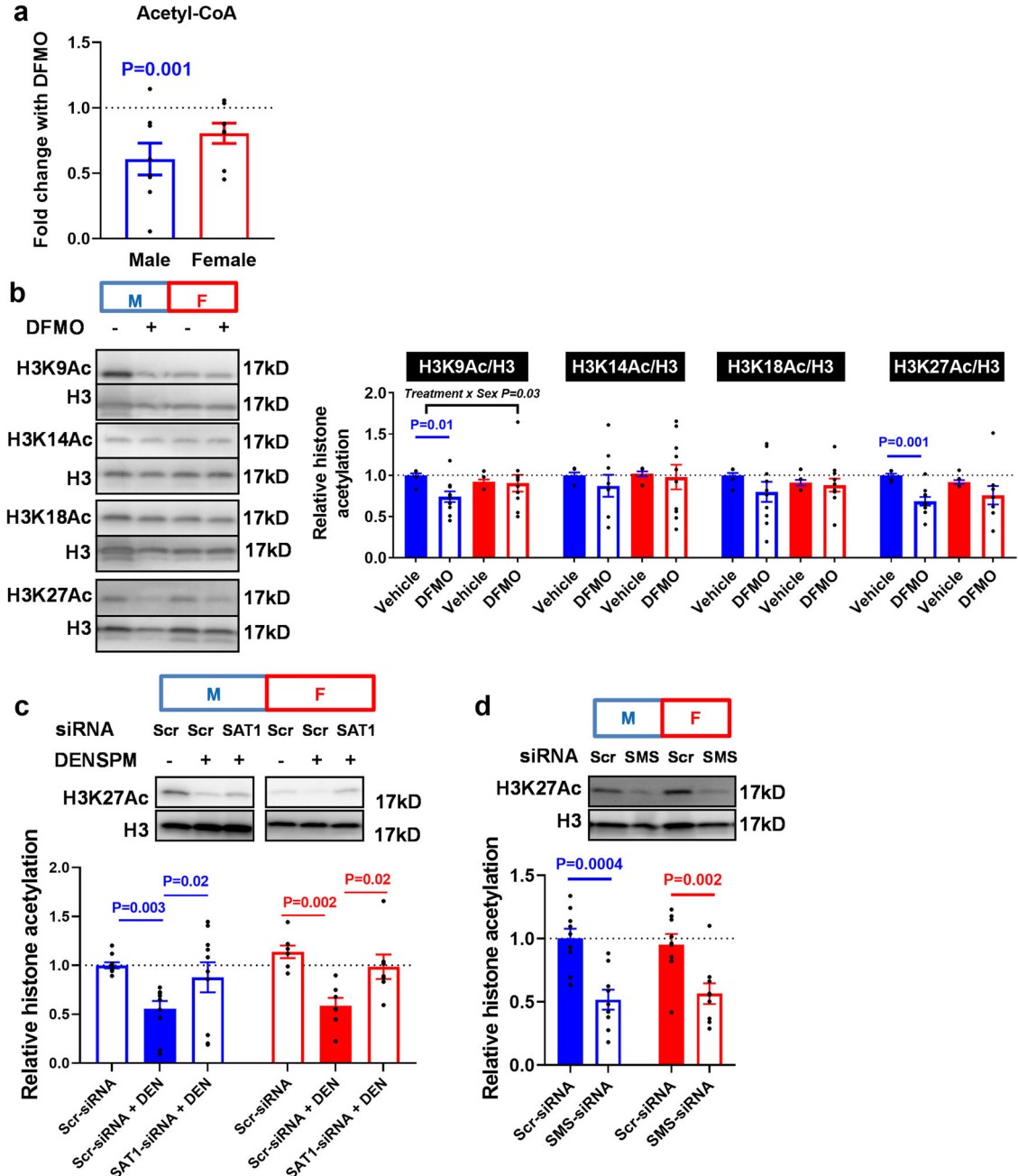

**Fig. 5 Polyamine depletion reduces acetyl-coA levels and decreases histone acetylation.** In male PHTs, DFMO decreases **a** acetyl-coA levels as measured by LC-MS and **b** histone acetylation as determined by immunoblotting. **c** H3K27Ac abundance is decreased by DENSPM (DEN) and reversed by SAT1 silencing. **d** H3K27Ac abundance is decreased following SMS-silencing. Bar graphs show mean ± SEM, $N = 8$–10 male PHTs and $N = 8$–10 female PHTs.

Polyamine depletion led to widespread changes in gene expression, consistent with a role of polyamines in gene regulation[28]. Pathway analyses indicated upregulation of inflammatory and stress signalling, and downregulation of mitochondrial metabolism genes following polyamine depletion. The inflammatory and stress response is consistent with previous reports that polyamines prevent stress[29–31]. However, the link to mitochondrial metabolism was intriguing as this has not been previously explored in detail. The polyamine end metabolites (i.e., catabolic end products of spermine) NAS and DAS were highly correlated with the TCA intermediates, and polyamine depletion decreased TCA intermediates as well as reduced OXPHOS activity. The TCA cycle and OXPHOS are coupled since the reciprocal generation of NADH/FADH2 and

NAD+/FAD respectively, fuels both processes[32]. Polyamines are mainly localized in the cytoplasm but rapid uptake of polyamines by the mitochondria has also been reported[33]. It is, therefore, possible that cellular polyamine depletion also led to reductions in mitochondrial polyamines which altered mitochondrial function.

Alterations in mitochondrial metabolites can influence the epigenetic landscape by, for example, regulating the availability of substrates for histone modification enzymes. The Michaelis constant (Km) of HATs fall within the range of acetyl-CoA concentrations typically found in cells[34]. Consequently, global levels of histone acetylation correlate with cellular acetyl-CoA abundance. Several findings from our study support a role for acetyl-CoA availability as the mechanism linking polyamine

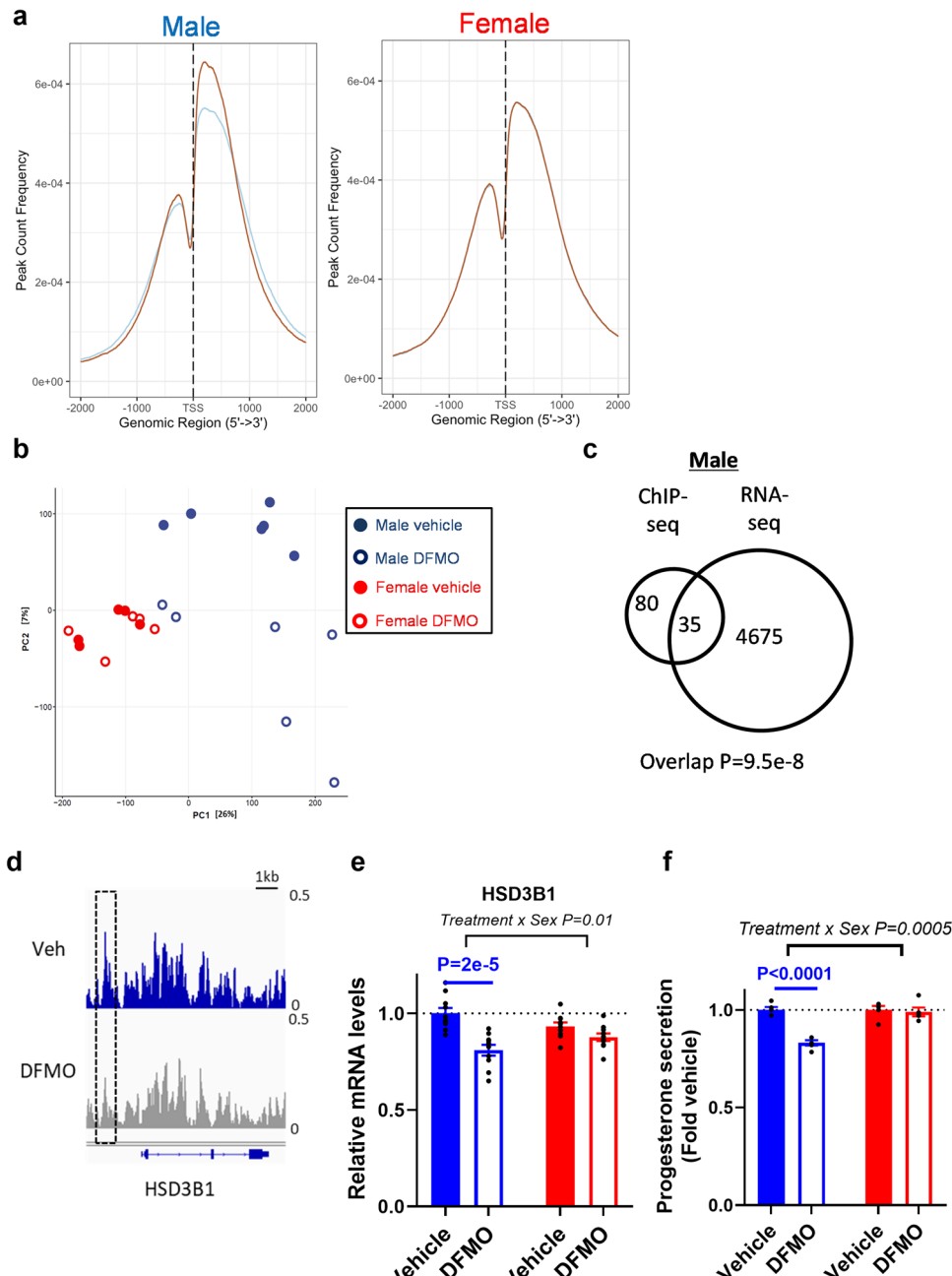

**Fig. 6 Polyamine depletion mediates global histone hypoacetylation and progesterone biosynthesis. a** H3K27Ac enrichment near transcription start sites is decreased in male PHTs following DFMO treatment. Orange line represents vehicle and blue line represents DFMO treated PHTs. **b** Unsupervised principal component analysis of H3K27Ac binding to genomic regions in vehicle and DFMO treated male and female PHTs. **c** Overlap of DBR annotated genes in H3K27Ac ChIP-seq analysis and DEG in RNA-seq analysis. **d** Custom views of differentially acetylated region in vehicle and DFMO treated male PHTs. Polyamine depletion decreased **e** HSD3B1 mRNA and **f** progesterone secretion in male PHTs. ChIP-seq experiments performed in $N = 6$ male PHTs and $N = 5$ female PHTs; progesterone secretion measured in $N = 5$ male PHTs and $N = 5$ female PHTs. Male and female samples are coloured blue and red respectively.

metabolism to histone acetylation. Glycolytic activity and acetyl-CoA levels were decreased in polyamine depleted PHTs, as were several histone acetylation marks, indicating global histone hypoacetylation. Furthermore, ChIP-seq analysis of the most robustly decreased histone acetylation mark (H3K27Ac) showed predominantly decreased acetylation in numerous loci, consistent with a global hypoacetylation event. It has been shown that induction of progressive mtDNA depletion which causes mitochondrial dysfunction results in decreased acetyl-CoA levels and global histone hypoacetylation[35]. Overall, a picture emerges

whereby polyamines have specific effects on the mitochondria that affect the histone acetylation landscape but determining how this occurs requires further study.

Mechanistically, our data suggest that polyamine-mediated regulation of gene expression is determined, at least in part, by regulating acetyl-CoA availability which is necessary for histone acetylation. In future studies, it will be informative to explore the relationship between polyamines and central energy metabolism, and their link to epigenetic regulation. Although histone acetylation explains some of the effects of polyamine depletion on

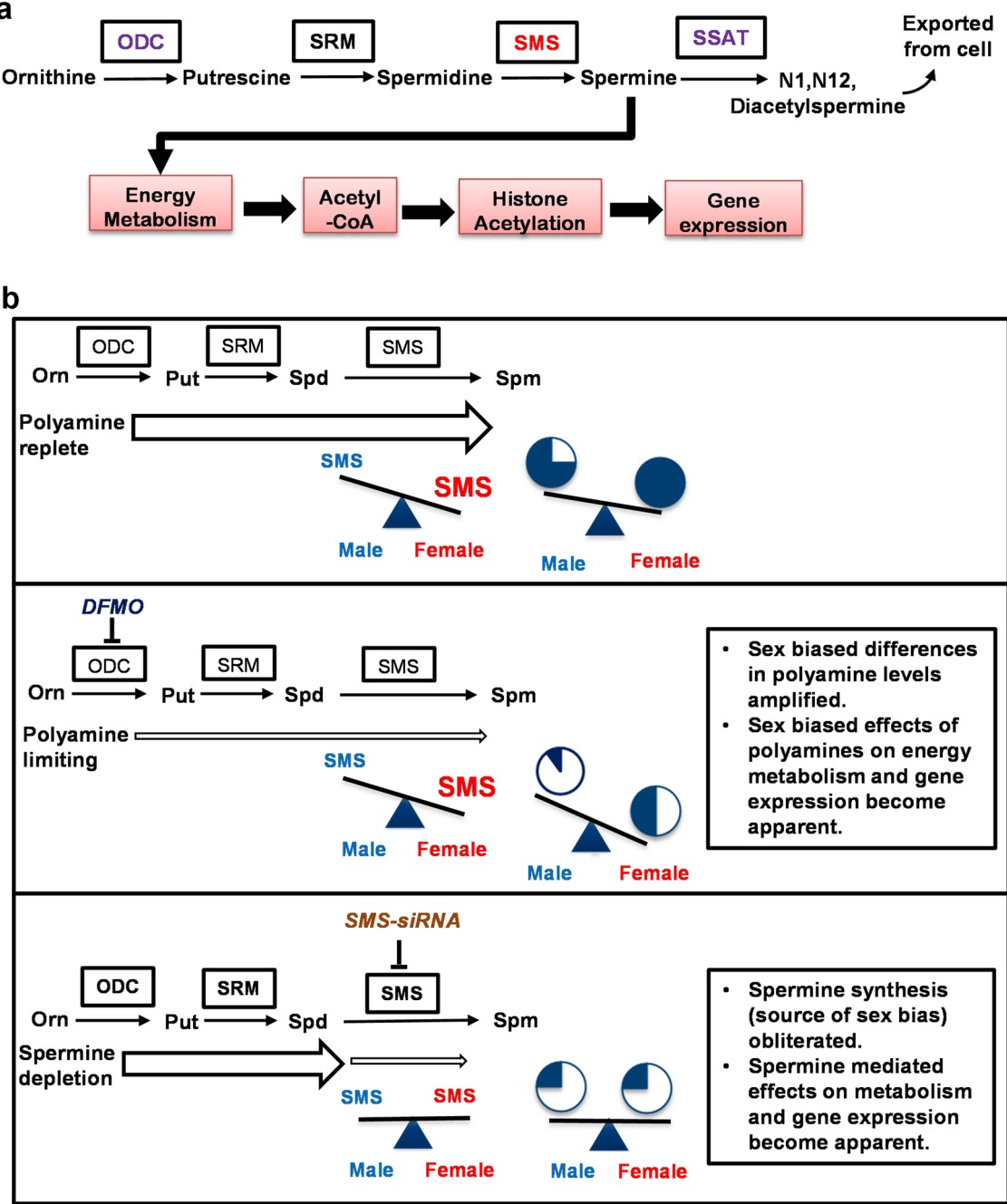

**Fig. 7 Schematic overview of polyamine metabolism and the mechanism of polyamine-mediated gene expression. a** Polyamine metabolic pathway with key enzymes highlighted and the downstream effects on trophoblast function. Spermine regulates cellular energy metabolism to increase acetyl-coA availability for histone acetylation and gene expression. Inhibition of ODC with DFMO, silencing of SMS via siRNAs, or activating SSAT using DENSPM decreases spermine levels leading to dysregulation of energy metabolism, histone acetylation, and gene expression. **b** Model of the effects of sex-biased polyamine metabolism on trophoblast function. Top panel: under physiological conditions, the female-biased SMS expression results in modest changes to spermine synthesis which are unlikely to result in sex-biased effects of polyamines on trophoblast function. Middle panel: Under polyamine limiting conditions the female bias in spermine synthesis buffers the effects of polyamine depletion leading to marked sex differences in spermine levels and sex-specific regulation of trophoblast function. Bottom panel: Upon spermine depletion, the effects on metabolism and gene expression become apparent but the sex-dependent effects are not observed because the source of the sex-bias is no longer present. Spermine levels are represented by the pie-charts.

gene regulation, additional factors are likely involved given a large number of DEGs that were not accounted for by histone acetylation. Other epigenetic effects such as histone methylation by α-ketoglutarate-dependent dioxygenases are plausible mechanisms linking polyamine-mediated mitochondrial dysfunction to histone modifications. Moreover, additional DNA-protein binding interactions also cannot be ruled out.

The locus-specific H3K27Ac and gene expression changes driven by polyamine depletion resulted in physiological outcomes as demonstrated by decreased acetylation and expression of HSD3B1, the enzyme critical for progesterone synthesis in the placenta, which is the main source of progesterone after 10 weeks gestation[36]. Progesterone is essential for maintaining pregnancy and low levels of progesterone lead to miscarriage. Our findings

are consistent with a previous study demonstrating H3K27 acetylation of HSD3B1 in PHTs[37]. Moreover, the reduced progesterone secretion by polyamine depletion in PHTs are also consistent with findings in mice showing a marked fall in progesterone following DFMO treatment[8]. Therefore, our study establishes the mechanism linking polyamines and progesterone production by demonstrating that polyamines promote H3K27 acetylation and transcription of HSD3B1 leading to increased progesterone synthesis.

In conclusion, our study demonstrates that polyamine metabolism influences trophoblast gene expression by regulating acetyl-coA synthesis and histone acetylation. The sex-differences in the effects of polyamine depletion can be explained by XCI escape of SMS leading to greater spermine synthesis in female trophoblasts and the downstream effects of spermine on acetyl-coA metabolism, histone acetylation, and gene expression. Taken together with our previous report[3], these findings suggest that the dysregulated polyamine metabolism in pregnancy complications may be linked to impaired placental function due to altered trophoblast histone acetylation and gene misregulation.

# Methods
Please refer to Supplemental Data 6 for the complete list of key resources.

**Placental tissue collection**. Placental tissues for primary human trophoblast cultures were collected from healthy women with normal term pregnancies and scheduled for delivery by elective Cesarean section. Participants were consented for research sample collection as part of the surgical procedure, with further permission for storage and transfer of materials to the biobank given under REC ID 07/MRE05/44. Analysis was performed as part of the Cambridge Blood and Stem Cell Biobank REC ID 18/EE/0199.

Metabolite analysis was performed in 110 healthy normal term placentas from the Pregnancy Outcome Prediction (POP) Study, previously described in detail[38,39]. Ethical approval for the study was given by the Cambridgeshire 2 Research Ethics Committee (REC ID 07/H0308/163) and all participants provided written informed consent.

**Primary human trophoblast (PHT) culture and targeting of polyamine metabolism**. A total of 104 placentas were processed for primary human trophoblast (PHT) culture. PHTs were isolated by trypsin digestion and Percoll purification as previously described[40,41]. Briefly, approximately 40 g of villous tissue was digested in trypsin (0.25%, Gibco) and DNAse I (325 Kunits/mg tissue, Sigma) and purified over a discontinuous 10–70% Percoll gradient centrifugation. Cells that migrated between 35 and 55% Percoll layers were collected and cultured in 1:1 mixture of Dulbecco's modified Eagle's medium (Sigma-Aldrich) and Ham's F-12 nutrient mixture (Gibco) containing 10% foetal bovine serum, 50 µg/ml gentamicin, 60 µg/ml benzyl penicillin and 100 µg/ml streptomycin (Sigma), and incubated in a 5% $CO_2$ humidified atmosphere at 37 °C. Following 18 h of culture, attached cells were washed twice in warmed Dulbecco's PBS, and culture media was changed daily.

Pharmacological targeting of polyamine metabolism was achieved either by treatment with difluoromethylornithine (DFMO, 5 mM) or $N^1,N^{11}$-Diethylnorspermine (DENSPM, 10 µM) from 66 to 90 h of culture. For siRNA-targeting, PHTs were transfected with 100 nM of siRNAs targeting SMS or SAT1 using Dharmafect2 transfection reagent at 18 h of culture. All experimental analyses were performed at 90 h of culture.

**Metabolic phenotyping with Seahorse Bioanalyzer**. To measure mitochondrial oxygen consumption and extracellular acidification rates, a Seahorse XF96 analyzer (Agilent) was used for the Mito Stress Test according to the kit's instructions. Briefly, isolated PHTs were plated onto a Seahorse XFe96 microplate at $0.5 \times 10^6$ cells/well. PHTs were maintained in culture as described above until 90 h when OCR and ECAR assays were performed. One hour before the assay, the medium was changed to XF base Media (Seahorse Bioscience, Agilent) supplemented with 17.5 mM glucose, 1 mM sodium pyruvate and 2 mM L-glutamine (equivalent to the concentrations present in PHT culture media). Cell metabolic rates were measured using XF96 Extracellular Flux Analyzer. Oligomycin (2 µM), carbonyl cyanide p-trifluoromethoxyphenylhydrazone (FCCP, 2 µM) and rotenone + antimycin (0.5 µM each) were sequentially added according to the experimental protocol[42]. Each experimental condition was repeated in 8 wells (technical replicates) per plate for each placenta, and the mean of the technical replicates considered to be an independent replicate.

**Metabolite measurements**. PHTs grown in 10 cm dishes at 10 million cells per dish, were used for the measurement of polyamine and TCA cycle metabolites. At the end of the experimental manipulations as described above, culture dishes were placed on ice, washed twice in D-PBS, and cells pelleted. For polyamine analyses, cell pellets were spiked with spermidine-(butyl-d8) trihydrochloride (Sigma Aldrich, #709891) as an internal standard and extracted in pre-chilled (−20 °C) acetonitrile. Following centrifugation to remove insoluble material, extracts were dried under nitrogen gas and reconstituted in 0.1% formic acid in $H_2O$. For analysis of TCA intermediates, cell pellets were extracted in methanol and spiked with internal standards, and dried down in a rotary evaporator. Extracts were then reconstituted in 10 mM ammonium acetate in $H_2O$. For placental biopsies, approximately 10 mg of frozen tissue was homogenized on a Bioprep-24-1004 homogenizer prior to metabolite extraction as described above.

LC-MS was performed as previously described[3]. Briefly, chromatographic separation was achieved on a ACE Excel 2 C18-PFP (150 mm * 2.1 mm, 2 µm) LC-column with a Shimadzu UPLC system. The column was maintained at 55 °C with a flow rate of 0.5 ml/min. A binary mobile phase system was used with mobile phase A; 0.1% formic acid in water, and mobile phase B; 0.1% formic acid in acetonitrile. The gradient profile was as follows; at 0 min_0% mobile phase B, at 2.5 min_0% mobile phase B, at 5 min_100% mobile phase B, at 7.5 min_100% mobile phase B, at 7.6 min_0% mobile phase B, at 11 min_0% mobile phase B.

Mass spectrometry detection was performed on an Exactive orbitrap mass spectrometer operating in positive ion mode. Heated electrospray source was used, the sheath gas was set to 40 (arbitrary units), the aux gas set to 15 (arbitrary units), and the capillary temperature set to 250 °C. The instrument was operated in full scan mode at 4 Hz from m/z 75–500 Da.

**Detection of biallelic SMS mRNA expression in single nucleus female placentas**. We used the GATK pipeline[43] to identify hetSNPs (heterozygous Single Nucleotide Polymorphisms) in exonic regions of SMS from the RNA-seq alignment data (i.e., BAM files) of our human placenta cohort[11]. Briefly, the pipeline takes the following steps: (1) marking duplicate reads using 'markDuplicate' of Picard, (2) splitting reads that contain 'N's in their CIGAR string using 'splitN-Read' of GATK (subsequent submodules from GATK hereafter), (3) realignment of reads around the indel using 'IndelRealigner', (4) recalibrating base quality using 'BaseRecalibrator', (5) calling the variants using 'HaplotypeCaller', and finally (6) counting reads by the reference and alternative bases of hetSNPs using 'ASEReadCounter'. We also selected a X-linked gene that is subject to XCI (i.e., a negative control of SMS) based on the following conditions: (1) reasonable expression level (normalized read count >100), (2) no significant change of transcript abundance by sex ($P_{adj} > 0.4$ and log2(Fold Change)< 1.1), and (3) fulfilling aforementioned conditions for the placenta and 19 non-placental tissues from our previous study[3]. There were eight such X-linked genes and we chose TIMM17B as a pre-designed Taqman SNP genotyping assay for this SNP was readily available. We identified four female placentas that were heterozygous for the SMS SNP (position X:21940733A/G; dbSNP ID: rs34507903) and the TIMM17B SNP (position X:48894188G/A; dbSNP ID: rs1128363).

Nuclei from the corresponding frozen placental tissues were isolated using EZ prep buffer (Sigma #NUC-101). Frozen placental biopsies (25 mg) were homogenized using a 2 ml Kimble dounce tissue grinder and nuclei pellets resuspended in Nuclei Suspension Buffer (Clontech #2313A) before filtering through a 40 µm cell strainer. The collected nuclei were then stained with 10 µg/ml DAPI and diluted to 10,000 nuclei per ml in PBS (without Ca and Mg). Nuclei were sorted by FACS into single nucleus per well in a 96 well plate filled with nuclear lysis buffer. The single cell-to-Ct kit (ThermoFisher #4458327) was used to perform reverse transcription, cDNA preamplification using allele-specific primers, and cDNA synthesis. Genotyping by qPCR was performed using pre-designed Taqman SNP genotyping assay for TIMM17B rs1128363 (assay ID:C__11611029_1_) and custom Taqman SNP genotyping assay for SMS rs34507903 and contain sequence-specific primers and VIC and FAM dye-labelled probes to allow for allelic discrimination.

The Taqman SNP genotyper software was used to determine the calling of monoallelic (i.e., reference or alternative alleles) or biallelic (reference and alternative alleles) in isolated single nuclei using the autocalling method according to the manufacturer's instructions. In this function, the determining factor influencing the call is the "Quality value" that the software algorithm assigns to each data point. The quality value considers the angle (a measure of cluster separation) and the amplitude (a measure of the signal intensity) of the samples from the non-template control. The default quality value of 0.95 (95%) was set for these analyses and samples below this value were assigned as "undetermined" (indicated by x on the allelic discrimination plots).

**RNA-sequencing analysis**. Total RNA was isolated from PHT lysates and genomic DNA removed using the RNeasy Plus Mini Kit (Qiagen). RNA quality was assessed using the RNA Nano kit on an Agilent 2100 Bioanalyzer (Agilent Technologies), and RNA samples with an RNA integrity number >8 were deemed suitable for RNA-seq experiments. Total RNA libraries were prepared into an indexed library using TruSeq Stranded total RNA library prep kit (Illumina) according to the manufacturer's instructions. cDNA libraries were validated using high sensitivity DNA chip on the Agilent 2100 Bioanalyzer and quantified using

the KAPA Library Quantification Kit (Roche). Indexed libraries were pooled and sequenced with a 50 bp single-end protocol on a HiSeq4000 platform (Illumina) by the Cancer Research UK Cambridge Institute Genomics Core Facility.

The primer sequences and poor-quality bases were trimmed from the sequencing reads using cutadapt[44] after assessing the quality of reads using FastQC (v0.11.4). Transcript quantification was performed using Salmon (v0.9.1) against Ensembl 90 version (GRCh38) of transcript annotations. Relative transcript abundance was measured in RPKM (Read Per Kilobase of transcript per Million mapped reads) and differentially expressed gene analysis was performed using DESeq2 (v1.18.1) Bioconductor package[45].

Pathway analysis was performed by gene set enrichment analysis (GSEA v.4.1.0) which uses predefined gene sets from the Molecular Signatures Database (MSigDB v7.4). Hallmark gene sets and C2 curated KEGG subset gene sets were used and list of ranked genes based on FPKM values. Enriched pathways with normalized enrichment scores <0.25 and false discovery rate <5% were considered significant.

**Chromatin immunoprecipitation (ChIP) Sequencing and ChIP-qPCR**. PHTs grown in 10 cm dishes (10 million cells per dish) and treated with DFMO or vehicle, were cross-linked in 1% formaldehyde in PBS for 10 min. Cross-linking was terminated by adding 125 mM glycine for 5 min, rinsed twice in cold D-PBS. Cells were then flash frozen as cell pellets and stored at −80 °C. Chromatin shearing were performed using Diagenode Chromatin shearing kit for Histones (#C01020010) with modifications. Briefly, cell pellets were re-suspended in Lysis Buffer 1 and homogenized on a dounce homogenizer to release nuclei and pelleted by centrifugation. Dounce homogenization was repeated using Lysis Buffer 2 and released nuclei pelleted by centrifugation. Nuclei were then lysed in iS Buffer 1, and chromatin was fragmented to approximately 200 bp by sonication for 6 cycles of 30 seconds on and 30 s off, on a Diagenode Bioruptor Pico. A small aliquot (20 μl) of sheared chromatin fragments were RNAseA and proteinase K digested, DNA purified on columns and the DNA size distribution examined using the Agilent 4200 TapeStation (Agilent Technologies). Chromatin concentrations were then quantified using a Qubit fluorometer prior to immunoprecipitation. The remaining sonicated chromatin (2 μg) was immunoprecipitated with 10 μg of H3K27Ac antibody (Abcam ab4729) or normal rabbit IgG in ChIP Buffer (CST) and incubated on a rotator at 4 °C overnight, or 10% of chromatin set aside as the input fraction. ChIP-grade Protein G magnetic beads (CST) were then added to the mixture and incubated for a further 4 h. The antibody-bound beads were washed, cross-linking was reversed and DNA fragments purified using QIAquick DNA purification kit (Qiagen).

ChIP-seq libraries were prepared on 5 ng of DNA from ChIP and input samples using the SMARTer ThruPlex DNA-seq kit along with DNA HT Dual Index Kit (both from Takara Bio) according the manufacturer's instructions. Libraries were sequenced on a HiSeq4000 (Illumina) with single end reads of 50 bp in the Cancer Research UK Cambridge Institute Genomics Core Facility. We obtained a median of 15 M reads across 43 samples (22 ChIP and 21 input samples). The primer sequences and poor quality bases were trimmed from the sequencing reads using cutadapt (v2.7 with Python 3.6.8)[44]. Trimmed reads were mapped to GRCh38 version human genome using bowtie (v1.1.2) and duplicated reads were discarded using sambamba (v0.7.1). On average, 72% of reads were uniquely mapped and they were used for subsequent analyses. The alignment files of input samples (i.e., DNA input without ChIP) were merged into four groups by taking the following two factors: (1) treatment types (DFMO or vehicle), and (2) the sex of PHTs. Then we identified peak regions of ChIP samples using MACS2 (v2.2.7.1) by matching the treatment type and the sex between ChIP and input samples to remove background signals. We used 148 bp as the fragment size (–extsize) and $2.7 \times 10^9$ as the effective genome size (–gsize). A peak is considered significant if $P_{adj} < 0.05$. To increase specificity, peaks supported by at least two ChIP samples of the same treatment-sex group were used and they were finally merged across the four groups. A total of 71,231 consensus peaks were generated and they were annotated using ChIPseeker (v1.26). Finally, differential binding regions (DBRs) were detected, separately for male and female PHTs, using DiffBind (v3.0.13). We used the 'background' normalization by binning 15,000 bp across the genome and applied the normalization factor to the full library size. We used a paired design with the following formular, which was subsequently used by DESeq2 and edgeR: ~pair + treatment.

Results from the ChIP-seq analyses were validated in biological replicates using ChIP-qPCR. Following ChIP with H3K27Ac as described above, qPCR analysis was performed using primers targeted against DNA regions of the following genes: INSIG1, HSD3B1, SLCO2B1, TGM2, VAC14, and ZDHHC1, as these are either implicated in progesterone homoeostasis and/or fulfil the criteria described in the results. Primers were designed based on enriched DNA regions identified by ChIP-seq analysis (see Supplemental Data 5 for primer sequences). The fold enrichment values were normalized to 10% input.

**Western blotting**. PHTs were harvested in RIPA buffer (50 mM Tris HCl, pH = 7.4; 150 mM NaCl; 0.1% SDS; 0.5% Na-deoxycholate and 1% Triton X100) containing protease inhibitors and phosphatase inhibitor cocktail 1 and 2 (1:100, Sigma). Protein lysates were separated on a 4–15% TGX gel (Biorad) under reducing and denaturing conditions, followed by transfer onto PVDF membranes. Amido Black (Sigma) staining was performed and membranes imaged for

quantification of total protein normalization before blocking in 5% BSA in TBS-0.1%Tween (TBS-T). Membranes were subsequently incubated in primary antibodies at 4 °C. Following washes in TBS-T, membranes were incubated with peroxidase-conjugated secondary antibodies and visualized by enhanced chemiluminescence using Clarity Western ECL substrate (Bio-Rad). Target protein levels were normalized against total protein (as determined by Amido Black staining).

**Statistics and reproducibility**. Data are presented as scatter plots showing mean ±95% confidence interval, scatter plots showing individual values, or dot blots showing means+SEM. Sample sizes (indicated in figures) always refers to biological replicates (i.e., PHTs or placentas from individual pregnancies). Statistical analyses were performed using GraphPad Prism9. Normal distribution was confirmed by Shapiro–Wilk test and data log-transformed if required. Unless otherwise stated, two-way ANOVA with treatment and sex as the principal factors and treatment by sex as the interaction term. The effects of treatment within each sex were examined post hoc by using the two-stage linear step-up procedure of Benjamini, Krieger, and Yekutieli[46], and adjusted $P < 0.05$ was considered significant. Relationships between polyamine metabolites and TCA cycle intermediates, or between RNA-seq and qPCR analyses were determined using Pearson's correlation coefficients and corrected for multiple comparisons by two-stage linear step-up method of Benjamini, Krieger, and Yekutieli[46]. Statistical methods for RNA-seq and ChIP-seq analyses are described in their respective Methods sections.

**Reporting summary**. Further information on research design is available in the Nature Research Reporting Summary linked to this article.

## Data availability
The authors declare that all the data supporting the findings of this study are available within the article, Supplemental Information, and Supplemental Data. The source data behind the graphs in the paper are available in Supplemental Data 7. Uncropped gel images are included in Supplemental Data 8. The software used in this study are listed in the key resources table (Supplemental Data 6). The RNA-Seq and ChIP-Seq datasets generated in this study are available in the European Nucleotide Archive (ENA www.ebi.ac.uk/ena) under the accession number PRJEB45391.

## Code availability
The software used in this study are listed in the key resources table (Supplemental Data 6). The code processing ChIP-Seq data is available from https://github.com/sung/placenta-polyamine-2022.

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

## Acknowledgements

We are grateful to the participants in the POP study; Emma Cook, Katrina Holmes, and Josephine Gill for technical assistance. Funding: The work was supported by a Centre for Trophoblast Research Next Generation Fellowship to Irving Aye and the NIHR Cambridge Biomedical Research Centre (Women's Health theme). The views expressed are those of the authors and not necessarily those of the NIHR or the Department of Health and Social Care. Giulia Avellino is supported by a PhD Scholarship from the Centre for Trophoblast Research and the BRC. Giulia Avellino and Roberta Barbagallo were supported by the Erasmus Plus traineeship. Figure panel 1a was created using Biorender (Biorender.com).

## Author contributions

I.L.M.H.A., G.A., R.B., and F.G. conducted the experiments. S.G. performed bioinformatics analyses. B.J.J. and A.K. performed mass spectrometry experiments. I.L.M.H.A., D.S.C.J., and G.C.S.S. designed the study. I.L.M.H.A., F.G., A.J.M., D.S.C.J., and G.C.S.S. interpreted the data and wrote the manuscript. All authors approved the manuscript.

## Competing interests

The authors declare no competing interests.
