## [Peer Review File · Communications Biology]

Reviewers' comments:

Reviewer #1 (Remarks to the Author):

Brief summary of the manuscript

The present study showed that the enzyme spermine synthase (SMS) escapes X chromosome inactivation (XCI) in female placentas leading to higher expression, and higher levels of the polyamine, spermine (DAS). In primary trophoblasts (PHT), inhibiting an enzyme upstream of SMS (using DFMO) reduced spermine levels, with greater reduction in male PHT. This suggests that higher levels in SMS in females might mitigate the effects of DFMO. Inhibition with DFMO also altered genes associated with the tricarboxylic acid cycle (TCA) and oxidative phosphorylation, and so mitochondrial metabolism was investigated. DFMO also reduced TCA cycle intermediates, and reduced oxidative phosphorylation and glycolysis in males but not females, again consistent with a model whereby increased SMS expression in females can mitigate the effects of DFMO. Alternatives to DFMO to deplete polyamines produced consistent results, suggesting that the results seen with DFMO were not due to off-target effects.

Consistent with the results above, DFMO also reduced acetyl-CoA levels, and the levels of some histone acetylation marks in male but not female PHTs. In addition to this effect potentially mediated by histone acetyltransferases (HATs), the authors also investigated effects of deacetylation by histone deacetylases (HDACs) by inhibiting these enzymes. ChIP-seq analysis of one of the altered marks (H3K27Ac) showed that DFMO changed the histone mark in male PHT but not females, and some of the differential bound regions overlapped with changes in gene expression (described above), suggesting that the change in polyamine levels led to changes in TCA, which led to changes in histone acetylation, which led to changes in gene expression, including HSD3B1, an enzyme involved in placental progesterone synthesis. DFMO also reduced progesterone release in male PHTs but not female PHTs, consistent with the overall model.

Overall impression of the work

I hope that my summary conveys that this is a detailed and thorough study, and elegantly examines multiple steps from XCI escape in one gene to various potential functional effects. Numerous studies catalog sex differences in placental function and pregnancy outcome, but this is one of the first to provide a plausible, detailed mechanism underlying sex differences.

Specific comments, with recommendations for addressing each comment

1. As described above, this study investigates multiple steps between SMS and potential functional differences between males and females, and upon careful reading, it all makes sense. However, I think it would be useful for the casual reader to have a figure showing a model of how everything fits together (and why higher SMS levels in females would be expected to mitigate effects of DFMO). This would probably be similar to Supplemental Figure 5, but more detailed, showing all of the steps examined.

2. There are two broad findings of this work: potential roles of polyamines, and sex-dependent effects arising from SMS escape from XCI. The latter is based on testing PHT from males and females separately and finding significance in one but not the other. I generally believe that this approach is insufficient, and that it is necessary to test the statistical interaction between sex and treatment, to show that the effect of treatment is significantly different in one sex than in the other. In this case, I acknowledge that the patterns are consistent across multiple tests, and there is a biological rationale for expecting males to be affected more than females. However, in a few cases, the sex-dependence is less convincing, e.g., the pattern in Fig 4B does not look that different from Fig 4D, even though the former suggests sex dependence while the latter does not. Similarly, in Fig 5B, the pattern for H3K27Ac does not look that different in males than in females, even though it is significant in one but not the other. I suggest that the authors test for and report sex by treatment interactions, just to be robust. If they are not significant (and I doubt they will be), I don't think the authors will need to change much text; even if not significant, the patterns

are consistent across many different measures- the effects are bigger in males, even if they are not "specific" to males.

3. A statement in the last paragraph of the Discussion is a bit too speculative in my opinion ("which confers a protective effect against pregnancy complications"). The authors have not shown a direct link between the functions affected and pregnancy complications.

4. The reference to Trivers-Willard is, I believe, misplaced. This hypothesis pertains to parental investment, but in this case the effects are due to what is happening in the placenta (i.e., same genotype as the fetus). Why would the placenta collude with the mother, against the interests of the fetus? I.e., the mother may have an interest in investing less in a male fetus under certain conditions, but the male fetus doesn't. Conversely, a female fetus may derive an advantage of SMS escaping XCI (to increase its expression to buffer against suboptimal conditions), but if a female placenta can have SMS escape XCI, why couldn't a male placenta upregulate SMS when needed?

5. In Supplemental Figure 1A, there are some "undetermined alleles" (black x), that occur on the plot where the heterozygotes are expected. How were "undetermined" distinguished from heterozygotes, and how can we be confident that the "undetermined" are not in fact heterozygotes?

Minor

6. "We previously purported that XCI contributes" I think purported is the wrong word here.

7. Fig 1B. Is it possible to present these data in some other way (not pie charts) that illustrates uncertainty in the proportions (i.e., error bars), and the sample size of nuclei?

8. Supplemental Figure 4A needs to be explained a bit better, i.e., "Mitochondrial respiration modulators (oligomycin, FCCP, Rotenone/Antimycin A [Rot/AA]) were injected at the indicated time points" presumably refers to the vertical dotted lines.

9. "As shown in Supplemental Figure 5B, HDAC inhibition by TSA increased H3K27 acetylation that was partially reversed by DFMO treatment. This data suggests that the effects of polyamine depletion on histone acetylation are not mediated by HDAC activity, but rather by regulating acetyl-CoA availability." And "The histone acetylation effects were independent of HDAC activity ." Isn't the finding consistent with both HDAC and HAT activity playing a role?

10. Fig 6B. I can't figure out what this analysis did. What was correlated with what to generate the principal components?

11. "thus resulting in 115 common DBRs . By integrating with RNA-seq data, 35 (~30%) DBRs" These numbers are not consistent with those shown in Fig 6C.

12. Supplemental Figs 4 and 6. Where values are standardized to 1 for the controls, there should still be error bars.

13. "To the best of our knowledge, this is the first report that an XCI escapee (SMS) contributes to the female-bias in placental function"

See

<https://www.nature.com/articles/s41467-018-04992-1>

<https://www.pnas.org/content/110/13/5169>

and related work from Tracy Bale's lab. This work is mostly in mice, but OGT is also an XCI escapee in humans, and this work shows functional consequences for the fetus resulting from changes in placental function.

Reviewer #2 (Remarks to the Author):

This paper presents important new understanding about the role of X chromosome inactivation (or lack of) in sex specific differences in the trophoblast.

The paper presents a comprehensive set of experiments that show the bi-allelic expression of spermine synthase at the single cell level and the impacts of polyamine biosynthesis on gene expression and histone acetylation in a sex specific fashion.

However, there appears to be several big jumps in logic, and this has the effect of both making the logic of the paper difficult to follow and makes it difficult to see how the specific discoveries made link back to the overarching question posed by the paper, i.e. the role of spermine synthesis in sex specific differences in the trophoblast.

I have identified two major comments that the authors should address.

1) It is not clear how the DFMO treatment links to the differences in SMS between sexes. I realize that there is less SMS rna and DAS in males, but in any of your treatments did you find that DFMO treated female PFTs behaved more like male PFTs? Was any attempt to look at whether these treatments resulted in changes consistent with sex differences? If you can't explain the sex specific differences with DFMO treatment, then can you really attribute these observations as an explanation for the sex differences themselves.

2) This paper is very difficult to digest. This could be improved, by more obvious statements that link observations.

For example, it is not immediately obvious why you would believe there is a causal link between SMS mRNA and DAS. This link is certainly implied. Some reading tells me that SMS is responsible for polyamine metabolism, which should impact on downstream polyamine metabolites. I think that explaining these links in the body of the text would make the paper easier to follow.

The manuscript may also benefit from the inclusion of a cartoon that illustrates your model and links each of your experimental findings.

Reviewer #3 (Remarks to the Author):

This study provides clear evidence that escape from X chromosome inactivation of the spermine synthase (SMS) gene in female trophoblasts contributes increased spermine. The author further showed that spermine differentially regulates trophoblast function in association with different alterations in acetylation levels in female and male cells. The data is novel and may partially explain upregulated spermine production in female placentas. There are several shortfalls regarding mechanisms underlying sexual dimorphism in spermine production in female and male trophoblasts, which needs to be clarified:

1. The escape from X chromosome inactivation of SMS gene in female trophoblasts was identified only in 4 female placentas (out of over 100 female placentas, assumingly based on Ref 3). Also as the rate of biallelic expression of SMS highly variable, ranging from 4 to 40% of individual nuclei. As such, it seems that the majority (over 60%?) of female trophoblast nuclei do not escape from X-inaction of SSM genes. In this case, it is questionable if this escape is not a major mechanism contributing increased spermine in female trophoblast. If so, what are the other mechanisms?
2. As spermine is believed to a key enhancer for OCR and glycolysis in trophoblast, it would be nice if the authors can discuss if and why female trophoblast cells have decreased OCR and glycolysis as compared to male cells (Fig. 4A), even though female cells produce more spermine than male cells.
3. As compared with the SMS siRNA, it appears that DFMO is not a potent inhibitor for spermine production in female trophoblast (Fig. 4A-D). Thus, it is highly recommended that the similar SMS

siRNA assays should be run to confirm the data shown in Fig. 5B and 6D-F as these data are important for the conclusions of this study.

Ref.: Ms. COMMSBIO-21-3079-T: *Placental sex-dependent spermine synthesis regulates trophoblast gene expression through acetyl-coA metabolism and histone acetylation*

Reviewers' comments:

Reviewer #1 (Remarks to the Author):

Brief summary of the manuscript

The present study showed that the enzyme spermine synthase (SMS) escapes X chromosome inactivation (XCI) in female placentas leading to higher expression, and higher levels of the polyamine, spermine (DAS). In primary trophoblasts (PHT), inhibiting an enzyme upstream of SMS (using DFMO) reduced spermine levels, with greater reduction in male PHT. This suggests that higher levels in SMS in females might mitigate the effects of DFMO. Inhibition with DFMO also altered genes associated with the tricarboxylic acid cycle (TCA) and oxidative phosphorylation, and so mitochondrial metabolism was investigated. DFMO also reduced TCA cycle intermediates, and reduced oxidative phosphorylation and glycolysis in males but not females, again consistent with a model whereby increased SMS expression in females can mitigate the effects of DFMO. Alternatives to DFMO to deplete polyamines produced consistent results, suggesting that the results seen with DFMO were not due to off-target effects.

Consistent with the results above, DFMO also reduced acetyl-CoA levels, and the levels of some histone acetylation marks in male but not female PHTs. In addition to this effect potentially mediated by histone acetyltransferases (HATs), the authors also investigated effects of deacetylation by histone deacetylases (HDACs) by inhibiting these enzymes. ChIP-seq analysis of one of the altered marks (H3K27Ac) showed that DFMO changed the histone mark in male PHT but not females, and some of the differential bound regions overlapped with changes in gene expression (described above), suggesting that the change in polyamine levels led to changes in TCA, which led to changes in histone acetylation, which led to changes in gene expression, including HSD3B1, an enzyme involved in placental progesterone synthesis. DFMO also reduced progesterone release in male PHTs but not female PHTs, consistent with the overall model.

Overall impression of the work

I hope that my summary conveys that this is a detailed and thorough study, and elegantly examines multiple steps from XCI escape in one gene to various potential functional effects. Numerous studies catalog sex differences in placental function and pregnancy outcome, but this is one of the first to provide a plausible, detailed mechanism underlying sex differences.

Specific comments, with recommendations for addressing each comment

1. As described above, this study investigates multiple steps between SMS and potential functional differences between males and females, and upon careful reading, it all makes sense. However, I think it would be useful for the casual reader to have a figure showing a model of how everything fits together (and why higher SMS levels in females would be expected to mitigate effects of DFMO). This would probably be similar to Supplemental Figure 5, but more detailed, showing all of the steps examined.

2. There are two broad findings of this work: potential roles of polyamines, and sex-dependent effects arising from SMS escape from XCI. The latter is based on testing PHT from males and females separately and finding significance in one but not the other. I generally believe that this approach is insufficient, and that it is necessary to test the statistical interaction between sex and treatment, to

show that the effect of treatment is significantly different in one sex than in the other. In this case, I acknowledge that the patterns are consistent across multiple tests, and there is a biological rationale for expecting males to be affected more than females. However, in a few cases, the sex-dependence is less convincing, e.g., the pattern in Fig 4B does not look that different from Fig 4D, even though the former suggests sex dependence while the latter does not. Similarly, in Fig 5B, the pattern for H3K27Ac does not look that different in males than in females, even though it is significant in one but not the other. I suggest that the authors test for and report sex by treatment interactions, just to be robust. If they are not significant (and I doubt they will be), I don't think the authors will need to change much text; even if not significant, the patterns are consistent across many different measures- the effects are bigger in males, even if they are not "specific" to males.

3. A statement in the last paragraph of the Discussion is a bit too speculative in my opinion ("which confers a protective effect against pregnancy complications"). The authors have not shown a direct link between the functions affected and pregnancy complications.

4. The reference to Trivers-Willard is, I believe, misplaced. This hypothesis pertains to parental investment, but in this case the effects are due to what is happening in the placenta (i.e., same genotype as the fetus). Why would the placenta collude with the mother, against the interests of the fetus? I.e., the mother may have an interest in investing less in a male fetus under certain conditions, but the male fetuses doesn't. Conversely, a female fetus may derive an advantage of SMS escaping XCI (to increase its expression to buffer against suboptimal conditions), but if a female placenta can have SMS escape XCI, why couldn't a male placenta upregulate SMS when needed?

5. In Supplemental Figure 1A, there are some "undetermined alleles" (black x), that occur on the plot where the heterozygotes are expected. How were "undetermined" distinguished from heterozygotes, and how can we be confident that the "undetermined" are not in fact heterozygotes?

Minor

6. "We previously purported that XCI contributes" I think purported is the wrong word here.

7. Fig 1B. Is it possible to present these data in some other way (not pie charts) that illustrates uncertainty in the proportions (i.e., error bars), and the sample size of nuclei?

8. Supplemental Figure 4A needs to be explained a bit better, i.e., "Mitochondrial respiration modulators (oligomycin, FCCP, Rotenone/Antimycin A [Rot/AA]) were injected at the indicated time points" presumably refers to the vertical dotted lines.

9. "As shown in Supplemental Figure 5B, HDAC inhibition by TSA increased H3K27 acetylation that was partially reversed by DFMO treatment. This data suggests that the effects of polyamine depletion on histone acetylation are not mediated by HDAC activity, but rather by regulating acetyl-CoA availability." And "The histone acetylation effects were independent of HDAC activity."

Isn't the finding consistent with both HDAC and HAT activity playing a role?

10. Fig 6B. I can't figure out what this analysis did. What was correlated with what to generate the principal components?

11. "thus resulting in 115 common DBRs . By integrating with RNA-seq data, 35 (~30%) DBRs" These numbers are not consistent with those shown in Fig 6C.

12. Supplemental Figs 4 and 6. Where values are standardized to 1 for the controls, there should still be error bars.

13. "To the best of our knowledge, this is the first report that an XCI escapee (SMS) contributes to the female-bias in placental function"

See

<https://www.nature.com/articles/s41467-018-04992-1>

<https://www.pnas.org/content/110/13/5169>

and related work from Tracy Bale's lab. This work is mostly in mice, but OGT is also an XCI escapee in humans, and this work shows functional consequences for the fetus resulting from changes in placental function.

Reviewer #2 (Remarks to the Author):

This paper presents important new understanding about the role of X chromosome inactivation (or lack of) in sex specific differences in the trophoblast.

The paper presents a comprehensive set of experiments that show the bi-allelic expression of spermine synthase at the single cell level and the impacts of polyamine biosynthesis on gene expression and histone acetylation in a sex specific fashion.

However, there appears to be several big jumps in logic, and this has the effect of both making the logic of the paper difficult to follow and makes it difficult to see how the specific discoveries made link back to the overarching question posed by the paper, i.e. the role of spermine synthesis in sex specific differences in the trophoblast.

I have identified two major comments that the authors should address.

1) It is not clear how the DFMO treatment links to the differences in SMS between sexes. I realize that there is less SMS rna and DAS in males, but in any of your treatments did you find that DFMO treated female PFTs behaved more like male PFTs? Was any attempt to look at whether these treatments resulted in changes consistent with sex differences? If you can't explain the sex specific differences with DFMO treatment, then can you really attribute these observations as an explanation for the sex differences themselves.

2) This paper is very difficult to digest. This could be improved, by more obvious statements that link observations.

For example, it is not immediately obvious why you would believe there is a causal link between SMS mRNA and DAS. This link is certainly implied. Some reading tells me that SMS is responsible for polyamine metabolism, which should impact on downstream polyamine metabolites. I think that explaining these links in the body of the text would make the paper easier to follow.

The manuscript may also benefit from the inclusion of a cartoon that illustrates your model and links each of your experimental findings.

Reviewer #3 (Remarks to the Author):

This study provides clear evidence that escape from X chromosome inactivation of the spermine synthase (SMS) gene in female trophoblasts contributes increased spermine. The author further showed that spermine differentially regulates trophoblast function in association with different alterations in acetylation levels in female and male cells. The data is novel and may partially explain upregulated spermine production in female placentas. There are several shortfalls regarding mechanisms underlying sexual dimorphism in spermine production in female and male trophoblasts, which needs to be clarified:

1. The escape from X chromosome inactivation of SMS gene in female trophoblasts was identified only in 4 female placentas (out of over 100 female placentas, assumingly based on Ref 3). Also as the rate of biallelic expression of SMS highly variable, ranging from 4 to 40% of individual nuclei. As such, it seems that the majority (over 60%?) of female trophoblast nuclei do not escape from X-inaction of SSM genes. In this case, it is questionable if this escape is not a major mechanism contributing increased spermine in female trophoblast. If so, what are the other mechanisms?
2. As spermine is believed to a key enhancer for OCR and glycolysis in trophoblast, it would be nice if the authors can discuss if and why female trophoblast cells have decreased OCR and glycolysis as compared to male cells (Fig. 4A), even though female cells produce more spermine than male cells.
3. As compared with the SMS siRNA, it appears that DFMO is not a potent inhibitor for spermine production in female trophoblast (Fig. 4A-D). Thus, it is highly recommended that the similar SMS siRNA assays should be run to confirm the data shown in Fig. 5B and 6D-F as these data are important for the conclusions of this study.

We thank the reviewers for their critical assessment, which has helped to improve the manuscript significantly. We have addressed all the concerns raised by the reviewers and revised the paper accordingly, as outlined below in red. We hope that the editor and the reviewers find these changes satisfactory.

REVIEWER 1

1. As described above, this study investigates multiple steps between SMS and potential functional differences between males and females, and upon careful reading, it all makes sense. However, I think it would be useful for the casual reader to have a figure showing a model of how everything fits together (and why higher SMS levels in females would be expected to mitigate effects of DFMO). This would probably be similar to Supplemental Figure 5, but more detailed, showing all of the steps examined.

Response: A detailed figure showing the polyamine metabolic pathway, experimental manipulations carried out and the summary of our findings and conclusions is now provided in Figure 7A and B. These figures are reproduced in the response to Reviewer 2 Comment 2.

Revisions made: Figure 7A shows the polyamine metabolic pathway and its effects on trophoblast function. Figure 7B illustrates the polyamine metabolic pathway highlighting the sex biased gene expression; targets of DFMO, siRNAs and their consequences on trophoblast function.

2. There are two broad findings of this work: potential roles of polyamines, and sex-dependent effects arising from SMS escape from XCI. The latter is based on testing PHT from males and females separately and finding significance in one but not the other. I generally believe that this approach is insufficient, and that it is necessary to test the statistical interaction between sex and treatment, to show that the effect of treatment is significantly different in one sex than in the other. In this case, I acknowledge that the patterns are consistent across multiple tests, and there is a biological rationale for expecting males to be affected more than females. However, in a few cases, the sex-dependence is less convincing, e.g., the pattern in Fig 4B does not look that different from Fig 4D, even though the former suggests sex dependence while the latter does not. Similarly, in Fig 5B, the pattern for H3K27Ac does not look that different in males than in females, even though it is significant in one but not the other. I suggest that the authors test for and report sex by treatment interactions, just to be robust. If they are not significant (and I doubt they will be), I don't think the authors will need to change much text; even if not significant, the patterns are consistent across many different measures- the effects are bigger in males, even if they are not "specific" to males.

Response: The results are now presented with statistical tests for interactions between sex and treatment. As the reviewer anticipated, not all of the observed treatment/sex interactions are statistically significant. Nevertheless, the direction of effect favoured a larger effect size in males and therefore our previous conclusions remain valid.

Revisions made: All figures where statistical interaction between sex and treatment was tested have now been revised. Specifically, Figure 4A-F, Figure 5A-D, Figure 6E-F, Supplemental Figure 5B-D, Supplemental Figure 6B, Supplemental Figure 7B, Supplemental Figure 8A-B. These revised graphs are not reproduced in this document as they constitute most of the graphs in this manuscript.

Additional revisions: Results: Lines 177-179; 264-266; 296-298, Methods: Lines 567-569.

3. A statement in the last paragraph of the Discussion is a bit too speculative in my opinion (“which confers a protective effect against pregnancy complications”). The authors have not shown a direct link between the functions affected and pregnancy complications.

Response: We agree with the reviewer that this may be too speculative. The last paragraph of the discussion has now been modified to reflect this.

Revisions made: The paragraph in question has been replaced with the following paragraph. Lines 364-377

In conclusion, our study demonstrates that polyamine metabolism influences trophoblast gene expression by regulating acetyl-coA synthesis and histone acetylation. The sex-differences in the effects of polyamine depletion can be explained by XCI escape of SMS leading to greater spermine synthesis in female trophoblasts and the downstream effects of spermine on acetyl-coA metabolism, histone acetylation and gene expression. Taken together with our previous report³, these findings suggest that the dysregulated polyamine metabolism in pregnancy complications may be linked to impaired placental function due to altered trophoblast histone acetylation and gene misregulation.

4. The reference to Trivers-Willard is, I believe, misplaced. This hypothesis pertains to parental investment, but in this case the effects are due to what is happening in the placenta (i.e., same genotype as the fetus). Why would the placenta collude with the mother, against the interests of the fetus? I.e., the mother may have an interest in investing less in a male fetus under certain conditions, but the male fetuses doesn't. Conversely, a female fetus may derive an advantage of SMS escaping XCI (to increase its expression to buffer against suboptimal conditions), but if a female placenta can have SMS escape XCI, why couldn't a male placenta upregulate SMS when needed?

Response: As above, this may also be too speculative. The last paragraph of the discussion has now been modified in light of this.

Revisions made: The paragraph in question has been replaced with the paragraph stated in the response to Reviewer 1 Comment 3 (reproduced above). Lines 364-377

5. In Supplemental Figure 1A, there are some “undetermined alleles” (black x), that occur on the plot where the heterozygotes are expected. How were “undetermined” distinguished from heterozygotes, and how can we be confident that the “undetermined” are not in fact heterozygotes?

Response: Signals were classified objectively. Specifically, we employed the Taqman SNP genotyper software to determine the calling of homozygotes or heterozygotes in isolated single nuclei using the autocalling method. In this function, the determining factor influencing the call is the “Quality value” that the software algorithm assigns to each data point. The quality value considers the angle (a measure of cluster separation) and the amplitude (a measure of the signal intensity) of the samples from the non-template control. The default quality value of 0.95 (95%) was set for these analyses and samples below this value were assigned as “undetermined”. Further detailed explanation is provided in the application note

<https://assets.thermofisher.com/TFS-Assets/GSD/Technical-Notes/quality-value-genotyper-software-technical-note.pdf>.

In our case where undetermined was called, there was a failure to amplify in either the reference, the alternative or both alleles which contributed to the lower-quality values. Given that we are performing these assays in single nuclei rather than bulk cells or tissues, the signals are inevitably low and prone to amplification failures (hence the substantial proportion of samples called as undetermined). Analysis was only performed when there is quantifiable signal/amplification and therefore we are confident that any calling (heterozygous or heterozygous) is based only on samples with significant signal.

Revisions made: The methods describing the analysis performed using the Taqman SNP genotyper software has been added. Lines 472-479.

The Taqman SNP genotyper software was used to determine the calling of monoallelic (i.e. reference or alternative alleles) or biallelic (reference and alternative alleles) in isolated single nuclei using the autocalling method according to the manufacturer's instructions. In this function, the determining factor influencing the call is the "Quality value" that the software algorithm assigns to each data point. The quality value considers the angle (a measure of cluster separation) and the amplitude (a measure of the signal intensity) of the samples from the non-template control. The default quality value of 0.95 (95%) was set for these analyses and samples below this value were assigned as "undetermined" (indicated by x on the allelic discrimination plots).

6. "We previously purported that XCI contributes" I think purported is the wrong word here.

Response: This has now been changed to "We previously hypothesized that XCI contributes".

Revisions made: Line 53-54.

7. Fig 1B. Is it possible to present these data in some other way (not pie charts) that illustrates uncertainty in the proportions (i.e., error bars), and the sample size of nuclei?

Response: This data provides qualitative evidence of biallelic SMS expression. Although percentage of biallelic expression is presented, this value is highly variable between placentas due to the limitations of the assays performed in single nuclei (see above response to Reviewer 1, Comment 5) and the heterogeneous nature of X chromosome inactivation escape. Because of the qualitative nature of these data, we believe that presentation of individual placentas is a better representation than the mean of the four placentas. Moreover, data on the individual nuclei (and sample sizes) from each placenta is provided in Supplemental Figure 1A. A more detailed explanation for interpretation of biallelic expression in single nuclei is also presented in the response to Reviewer 3 comment 1.

Revisions made: The results section has been expanded to explain the hetSNP prevalence and heterogeneity of XCI escape within each placenta. Lines 102-106; 116-120; 123-124

8. Supplemental Figure 4A needs to be explained a bit better, i.e., "Mitochondrial respiration modulators (oligomycin, FCCP, Rotenone/Antimycin A [Rot/AA]) were injected at the indicated time points" presumably refers to the vertical dotted lines.

Response: These have now been modified to include the mitochondrial respiration modulators in the graphs and the statement in question is now added to the figure legends.

Revisions made: The names of the mitochondrial respiration modulators have now been added to Supplemental Figure 4 and Supplemental Figure 4 legend: “Mitochondrial respiration modulators (oligomycin, FCCP, Rotenone/Antimycin A [Rot/AA]) were injected at the indicated time points shown by the dotted lines.”

9. “As shown in Supplemental Figure 5B, HDAC inhibition by TSA increased H3K27 acetylation that was partially reversed by DFMO treatment. This data suggests that the effects of polyamine depletion on histone acetylation are not mediated by HDAC activity, but rather by regulating acetyl-CoA availability.” And “The histone acetylation effects were independent of HDAC activity.”

Isn't the finding consistent with both HDAC and HAT activity playing a role?

Response: If the effects of polyamine depletion by DFMO were mediated by activation of HDACs (i.e. a mechanism independent of regulating acetyl-coA availability), inhibition of HDACs would completely prevent the DFMO response. Therefore, in Supplemental Figure 6B* there would be no difference between TSA vs DFMO+TSA. However, since there was a decrease in DFMO-mediated histone acetylation in the presence of HDAC inhibition (i.e. TSA), this suggests that HDACs do not play a clear role in mediating polyamine's effects. This has now been clarified.

*Please note that Supplemental Figure 5 is now Supplemental Figure 6 due to the addition of new experimental data and figures.

Revisions made: The following statement has been added (Lines 220-225):

We reasoned that if polyamine depletion by DFMO was mediated by HDACs, then inhibition of HDACs would prevent the DFMO-mediated changes in histone acetylation (Supplemental Figure 6A). As shown in Supplemental Figure 6B, HDAC inhibition by TSA increased H3K27 acetylation that was partially reversed by DFMO treatment, refuting this alternative hypothesis.

10. Fig 6B. I can't figure out what this analysis did. What was correlated with what to generate the principal components?

Response: The data used to generate the PCA plot was a matrix of the normalised ChIP-Seq read count for all consensus peak regions across all samples. We used PCA to then identify orthogonal distance and relatedness amongst fetal sex and treatment. The plot shows that male trophoblasts were clearly separated by DFMO and vehicle (separated by PC2 plotted on the y-axis), while female sample were not. We interpret these findings as follows: male trophoblasts were more sensitive to the effect of DFMO whereas females are more tolerant, suggesting a sex-based effect of DFMO treatment at the level of H3K27 acetylation level.

Revisions made: Figure 6B has now been updated (see below) with the labels on each sample removed so that the data is clearer.

Figure 6B,

The results describing Figure 6B has been expanded to clarify the rationale for this experiment and the findings (Lines: 234-239): *Unsupervised principal component analysis was performed to examine the relatedness in histone acetylation regions by DFMO treatment and fetal sex. This multivariate analysis shows that H3K27Ac peaks are clearly separated by treatment in male PHTs (i.e. vehicle vs DFMO) but lack this separation in female PHTs (Figure 6B), consistent with a sex-biased effect of polyamine depletion.*

11. "thus resulting in 115 common DBRs. By integrating with RNA-seq data, 35 (~30%) DBRs" These numbers are not consistent with those shown in Fig 6C.

Response: We apologise for this error. The numbers in the original Figure 6C refer to DBRs whereas the text refers to genes associated with the DBRs – in some cases different regions of the same gene were differentially acetylated. This has now been corrected to reflect only genes associated with the DBRs (i.e. 2 or more DBR in a single gene are counted as one).

Revisions made: Lines 249. *thus resulting in 146 common DBRs corresponding to 115 genes.*

12. Supplemental Figs 4 and 6. Where values are standardized to 1 for the controls, there should still be error bars.

Response: This has now been corrected.

Revisions made: Error bars for controls added to Supplemental Figure 5* and 7*

*Please note that Supplemental Figure 4 and 6 are now Supplemental Figures 5 and 7 due to the addition of new experimental data and figures.

13. “To the best of our knowledge, this is the first report that an XCI escapee (SMS) contributes to the female-bias in placental function”
See

<https://www.nature.com/articles/s41467-018-04992-1>

<https://www.pnas.org/content/110/13/5169>

and related work from Tracy Bale’s lab. This work is mostly in mice, but OGT is also an XCI escapee in humans, and this work shows functional consequences for the fetus resulting from changes in placental function.

Response: We have now removed this sentence in accordance with the journal’s formatting policy which states that claims of novelty should be avoided. We have also cited Dr Bale’s work on OGT.

Revisions made: Lines 271-272, 305-311

REVIEWER 2

1) It is not clear how the DFMO treatment links to the differences in SMS between sexes. I realize that there is less SMS rna and DAS in males, but in any of your treatments did you find that DFMO treated female PFTs behaved more like male PFTs? Was any attempt to look at whether these treatments resulted in changes consistent with sex differences? If you can’t explain the sex specific differences with DFMO treatment, then can you really attribute these observations as an explanation for the sex differences themselves.

Response: The sex-bias in spermine synthesis explains the differences in the effects of polyamine depletion between males and female trophoblasts. DFMO inhibits the biosynthesis of all polyamines but spermine production is higher in female PHTs due to the higher SMS expression in these tissues (Supplemental Figure 2A). Notably, when we targeted spermine metabolism (the source of the sex-bias) either by inhibiting spermine synthesis (SMS-silencing) or promoting spermine catabolism (SSAT-induction with DENSPM), we recapitulated the functional effects of polyamine depletion by DFMO, but these effects were observed in both males and females (Figure 4C-F; Figure 5C-D; Supplemental Figure 8A-B). This is because the source of the sex-bias (i.e. SMS expression and spermine metabolism) is no longer relevant under these conditions. This is supported by data showing that DFMO-treatment decreased spermine levels to a greater extent in male compared to female trophoblasts (Supplemental Figure 2A). However, SMS-silencing (which inhibits spermine synthesis) or SSAT-induction (which promotes spermine catabolism) decreased spermine in both males and females to a similar extent (Supplemental Figure 5C-D).

Revisions made: Additional experimental data showing that the effects of SMS-silencing recapitulate the key effects of DFMO but in both sexes has now been added. New figures added: Figure 5D, Supplemental Figures 5C-D, 8A-B (see response to Reviewer 3 Comment 3 for a reproduction of these figures in this document).

The discussion surrounding the explanation of DFMO-mediated sex differences has now been expanded including discussion of the additional data which clarifies this point. Lines 302-311.

Notably, depletion of spermine by SMS-silencing recapitulated many of the effects of polyamine depletion by DFMO but produced similar effects in both male and female PHTs. This can be explained due to the fact that the female bias in SMS expression is the source of the placental sex bias in polyamine metabolism. The polyamine metabolic pathway is highly regulated under physiological conditions, and therefore it is likely that the functional consequences of the sex-bias in spermine synthesis become apparent under polyamine limiting conditions (Figure 7B). Consistent with this hypothesis, previous studies demonstrate that the sex-specific effects of another placenta-specific XCI escapee, O-linked N-acetylglucosamine transferase (OGT), are unmasked by maternal stress.

2) This paper is very difficult to digest. This could be improved, by more obvious statements that link observations.

For example, it is not immediately obvious why you would believe there is a causal link between SMS mRNA and DAS. This link is certainly implied. Some reading tells me that SMS is responsible for polyamine metabolism, which should impact on downstream polyamine metabolites. I think that explaining these links in the body of the text would make the paper easier to follow. The manuscript may also benefit from the inclusion of a cartoon that illustrates your model and links each of your experimental findings.

Response: The polyamine metabolic pathway is now discussed in greater detail and a new Figure 7 has been added to illustrate the polyamine metabolic pathway, highlighting the enzymes targeted in this study, the source of the sex-bias and our interpretation of the key findings.

Revisions made: A new Figure 7 has been added and the figure legends summarise the conclusions and interpretation of the key findings in this study (reproduced below). Moreover, the manuscript has now been extensively modified throughout to simplify the text and provide obvious statements that link observations.

Lines: 66-72; 127-132; 160-162; 166-168; 188-189; 219-223; 271-272; 276; 296-298; 303-311; 317-319; 364-371.

Figure 7: Schematic overview of polyamine metabolism and the mechanism of polyamine-mediated gene expression.

(A) Polyamine metabolic pathway with key enzymes highlighted and the downstream effects on trophoblast function. Spermine regulates cellular energy metabolism to increase acetyl-coA availability for histone acetylation and gene expression. Inhibition of ODC with DFMO, silencing of SMS via siRNAs or activating SSAT using DENSPM decreases spermine levels leading to dysregulation of energy metabolism, histone acetylation and gene expression. (B) Model of the effects of sex-biased polyamine metabolism on trophoblast function. **Top panel:** under physiological conditions, the female-biased SMS expression results in modest changes to spermine synthesis which are unlikely to result in sex-biased effects of polyamines on trophoblast function. **Middle panel:** Under

polyamine limiting conditions the female bias in spermine synthesis buffers the effects of polyamine depletion leading to marked sex differences in spermine levels and sex-specific regulation of trophoblast function. **Bottom panel:** Upon spermine depletion, the effects on metabolism and gene expression become apparent but the sex-dependent effects are not observed because the source of the sex-bias is no longer present. Spermine levels are represented by the pie-charts.

REVIEWER 3

1. The escape from X chromosome inactivation of SMS gene in female trophoblasts was identified only in 4 female placentas (out of over 100 female placentas, assumingly based on Ref 3). Also as the rate of biallelic expression of SMS highly variable, ranging from 4 to 40% of individual nuclei. As such, it seems that the majority (over 60%?) of female trophoblast nuclei do not escape from X-inaction of SSM genes. In this case, it is questionable if this escape is not a major mechanism contributing increased spermine in female trophoblast. If so, what are the other mechanisms?

Response: This has been a mis-understanding of the biallelic expression analyses. To clarify the biallelic expression analyses, the goal of the SNP qPCR assays in single nuclei was to provide qualitative evidence of biallelic SMS mRNA expression. As the expression assay requires analysis of mRNA only exonic SNPs are informative. We only identified 4 placentas that were heterozygous for such a SNP and therefore it was only possible to perform the biallelic expression analyses in these 4 placentas. However, all 4 placentas containing this hetSNP showed biallelic SMS expression, therefore we confirmed XCI escape in 100% of the placentas analysed. The reason we only identified 4 placentas is because the minor allele frequency of the SMS hetSNP (rs34507903) in European populations is 0.26 or 26% (1000 genomes project), and TIMM17B hetSNP (rs1128363) is 0.47. Therefore, the chances of identifying placentas containing hetSNPs of both these genes from our total of 90 female placentas is $90 \times 0.26 \times 0.47 = \sim 11$ placentas. Moreover, although the mRNA level of SMS is higher in the placenta than any other tissues from GTEx consortium, the average sequencing depth for the selected hetSNP (X:21940733 A/G; <https://www.ncbi.nlm.nih.gov/snp/rs34507903>) is 6 reads. This is because the SNP is located within the 5' UTR region, where RNA-Seq often does not yield high coverage. Therefore, the prevalence of the hetSNP in the general population and the low sequencing coverage explains why only a small number of female placentas with hetSNPs in an exon region of SMS was identified and analysed.

With regards to the varying degree of biallelic expression of SMS in individual nuclei from these 4 female placentas, our findings are consistent with previous reports demonstrating a high level of variability in XCI escape between individual cells (ranging from 1% to 99% of single cells exhibiting biallelic expression)¹⁻⁵.

We have modified the results and discussion to explain the reasons for the prevalence of the hetSNP and clarify that the assays for biallelic SMS expression analyses in single nuclei are qualitative assays.

Revisions made: Lines 102-106; 116-120; 123-124

2. As spermine is believed to a key enhancer for OCR and glycolysis in trophoblast, it would be nice if the authors can discuss if and why female trophoblast cells have decreased OCR and glycolysis as compared to male cells (Fig. 4A), even though female cells produce more spermine than male cells.

Response: In Figure 4A the effects of sex on OXPHOS or glycolysis were not significantly different (OXPHOS P=0.4 and glycolysis P=0.09) but there was an interaction between sex and treatment in the OXPHOS response to DFMO. Regulation of cellular energy metabolism is complex and influenced by a vast array of biological processes, some of which can be regulated by fetal sex (e.g. polyamines).

Therefore we cannot conclude that the sex-differences in spermine under normal physiological conditions (i.e. without experimental manipulation) sufficiently influences OXPPOS or glycolysis; however, these sex-differences become apparent under conditions when polyamine metabolism is dysregulated.

Figure 7 has been added to illustrate the effects of polyamines and the role of sex-specific regulation of polyamine metabolism on placental function (please refer to the response to Reviewer 2, Comment 2 for a reproduction of this figure). The discussion has also been amended to discuss our hypothesis that the sex-specific effects only become apparent under polyamine limiting conditions.

Revisions made: Figure 7 added, and Discussions revised (Lines: 296-298; 302-311; 364-371).

3. As compared with the SMS siRNA, it appears that DFMO is not a potent inhibitor for spermine production in female trophoblast (Fig. 4A-D). Thus, it is highly recommended that the similar SMS siRNA assays should be run to confirm the data shown in Fig. 5B and 6D-F as these data are important for the conclusions of this study.

Response: The differences in OXPPOS activity following DFMO appears modestly higher than the effects observed with SMS-siRNA (Delta = 30 vs 18 pmol/min/mg respectively) whereas the differences in glycolytic activity are comparable (Delta = 14 vs 15 mpH/min/mg respectively). However, neither of these effects were statistically significant. Nevertheless, we now provide additional data demonstrating the effects of SMS-siRNA on polyamine levels, histone acetylation, HSD3B1 expression and progesterone secretion.

Revisions made: New figures added: Figure 5D (SMS-siRNA effect on H3K27 acetylation), Supplemental Figures 5C-D (SMS-siRNA effect on polyamine levels), Supplemental Figures 8A-B (SMS siRNA effects on HSD3B1 mRNA and progesterone secretion).

Figure 5D

Supplemental Figure 5C and D

Supplemental Figure 8A and B

References cited in this document:

1. Garieri, M. *et al.* Extensive cellular heterogeneity of X inactivation revealed by single-cell allele-specific expression in human fibroblasts. *Proceedings of the National Academy of Sciences of the United States of America* **115**, 13015–13020 (2018).
2. Carrel, L. & Willard, H. F. Heterogeneous gene expression from the inactive X chromosome: An X-linked gene that escapes X inactivation in some human cell lines but is inactivated in others. *Proceedings of the National Academy of Sciences* **96**, 7364–7369 (1999).
3. Wainer Katsir, K. & Linial, M. Human genes escaping X-inactivation revealed by single cell expression data. *BMC Genomics* **20**, 1–17 (2019).
4. Tukiainen, T. *et al.* Landscape of X chromosome inactivation across human tissues. *Nature* **550**, 244–248 (2017).
5. Phung, T. N., Olney, K. C., Kliman, H. J. & Wilson, M. A. Patchy, incomplete, and heterogeneous X-inactivation in the human placenta. *bioRxiv* (2020) doi:10.1101/785105.

REVIEWERS' COMMENTS:

Reviewer #1 (Remarks to the Author):

The authors have responded to my comments, and the manuscript has been improved. A few very minor errors were introduced in the revisions:

Line 68 reads "Spermidine and spermine can also be catabolized by the enzyme spermidine/spermine acetyltransferase (SSAT) into N1-acetyl-spermidine (NAS) and DAS respectively, which acetylates these metabolites facilitating their cellular export." "These metabolites" refers to spermidine and spermine, not NAS and DAS, so might be better phrased "Spermidine and spermine can also be acetylated by the enzyme spermidine/spermine acetyltransferase (SSAT) into N1-acetyl-spermidine (NAS) and DAS respectively, which facilitates their cellular export."

Line 127 "Compared to male PHTs, In female PHTs express 50% higher levels of SMS mRNA" Remove "In".

Line 235 "This multivariate analysis shows that H3K27Ac peaks were clearly separated by treatment in male PHTs (i.e. vehicle vs DFMO) but this separation in female PHTs (Figure 6B), consistent with a sex-biased effect of polyamine depletion."
Could be changed to "This multivariate analysis shows that H3K27Ac peaks were clearly separated by treatment in male PHTs (i.e. vehicle vs DFMO) but not in female PHTs (Figure 6B), consistent with a sex-biased effect of polyamine depletion."

Line 264 "Moreover, the interaction between sex and treatment were significantly different." should be "Moreover, the interaction between sex and treatment was significant."

Line 267 "global histone hypoacetylation mediated by polyamine depletion of polyamines " Delete the first "polyamine"

Line 393 "Briefly, approximately 40g of villous tissue [WAS] digested in trypsin (0.25%, Gibco) and DNase I (325 Kunits/mg tissue, Sigma) and purified over a discontinuous 10–70% Percoll gradient centrifugation." See missing word in ALL CAPS.

Line 568 "The effects of treatment within each sex were examined post-hoc by using the two-stage linear step-up procedure of Benjamini, Krieger and Yekutieli48 [AND] adjusted $P < 0.05$ was considered significant. See missing word in ALL CAPS.

Reviewer #3 (Remarks to the Author):

In this revision, the authors have addressed all of the major concerns this reviewer had. I have no further comments.

Ref.: Ms. COMMSBIO-21-3079-T: *Placental sex-dependent spermine synthesis regulates trophoblast gene expression through acetyl-coA metabolism and histone acetylation*

Second review

Reviewers' comments:

REVIEWERS' COMMENTS:

Reviewer #1 (Remarks to the Author):

The authors have responded to my comments, and the manuscript has been improved. A few very minor errors were introduced in the revisions:

1. Line 68 reads "Spermidine and spermine can also be catabolized by the enzyme spermidine/spermine acetyltransferase (SSAT) into N1-acetyl-spermidine (NAS) and DAS respectively, which acetylates these metabolites facilitating their cellular export." "These metabolites" refers to spermidine and spermine, not NAS and DAS, so might be better phrased "Spermidine and spermine can also be acetylated by the enzyme spermidine/spermine acetyltransferase (SSAT) into N1-acetyl-spermidine (NAS) and DAS respectively, which facilitates their cellular export."
2. Line 127 "Compared to male PHTs, In female PHTs express 50% higher levels of SMS mRNA" Remove "In".
3. Line 235 "This multivariate analysis shows that H3K27Ac peaks were clearly separated by treatment in male PHTs (i.e. vehicle vs DFMO) but this separation in female PHTs (Figure 6B), consistent with a sex-biased effect of polyamine depletion."
Could be changed to "This multivariate analysis shows that H3K27Ac peaks were clearly separated by treatment in male PHTs (i.e. vehicle vs DFMO) but not in female PHTs (Figure 6B), consistent with a sex-biased effect of polyamine depletion."
4. Line 264 "Moreover, the interaction between sex and treatment were significantly different." should be "Moreover, the interaction between sex and treatment was significant."
5. Line 267 "global histone hypoacetylation mediated by polyamine depletion of polyamines "
Delete the first "polyamine"
6. Line 393 "Briefly, approximately 40g of villous tissue [WAS] digested in trypsin (0.25%, Gibco) and DNase I (325 Kunits/mg tissue, Sigma) and purified over a discontinuous 10–70% Percoll gradient centrifugation." See missing word in ALL CAPS.
7. Line 568 "The effects of treatment within each sex were examined post-hoc by using the two-stage linear step-up procedure of Benjamini, Krieger and Yekutieli48 [AND] adjusted P<0.05 was considered significant. See missing word in ALL CAPS.

Reviewer #3 (Remarks to the Author):

In this revision, the authors have addressed all of the major concerns this reviewer had. I have no further comments.

We thank the reviewers for their further assessment of our manuscript. We have now addressed all the concerns which are summarised below.

Reviewer 1

1. Line 68 reads "Spermidine and spermine can also be catabolized by the enzyme spermidine/spermine acetyltransferase (SSAT) into N1-acetyl-spermidine (NAS) and DAS respectively, which acetylates these metabolites facilitating their cellular export." "These metabolites" refers to spermidine and spermine, not NAS and DAS, so might be better phrased "Spermidine and spermine can also be acetylated by the enzyme spermidine/spermine acetyltransferase (SSAT) into N1-acetyl-spermidine (NAS) and DAS respectively, which facilitates their cellular export."

Response: This has now been corrected

Revisions made: Line 63-65; Spermidine and spermine can also be acetylated by the enzyme spermidine/spermine acetyltransferase (SSAT) into N1-acetyl-spermidine (NAS) and DAS respectively, which facilitates their cellular export.

2. Line 127 "Compared to male PHTs, In female PHTs express 50% higher levels of SMS mRNA"
Remove "In".

Response: This has now been corrected

Revisions made: Line 122, Compared to male PHTs, female PHTs express 50% higher levels of SMS mRNA

Line 235 "This multivariate analysis shows that H3K27Ac peaks were clearly separated by treatment in male PHTs (i.e. vehicle vs DFMO) but this separation in female PHTs (Figure 6B), consistent with a sex-biased effect of polyamine depletion."

Could be changed to "This multivariate analysis shows that H3K27Ac peaks were clearly separated by treatment in male PHTs (i.e. vehicle vs DFMO) but not in female PHTs (Figure 6B), consistent with a sex-biased effect of polyamine depletion.

Response: This has now been corrected

Revisions made: Line 221-223, This multivariate analysis shows that H3K27Ac peaks were clearly separated by treatment in male PHTs (i.e. vehicle vs DFMO) but not in female PHTs (Figure 6B), consistent with a sex-biased effect of polyamine depletion

3. Line 264 "Moreover, the interaction between sex and treatment were significantly different."
should be "Moreover, the interaction between sex and treatment was significant."

Response: This has now been corrected

Revisions made: Line 245-246, Moreover, the interaction between sex and treatment was significant

4. Line 267 "global histone hypoacetylation mediated by polyamine depletion of polyamines "
Delete the first "polyamine"

Response: This has now been corrected

Revisions made: Line 248, global histone hypoacetylation mediated by depletion of polyamines

5. Line 393 "Briefly, approximately 40g of villous tissue [WAS] digested in trypsin (0.25%, Gibco) and DNase I (325 Kunits/mg tissue, Sigma) and purified over a discontinuous 10–70% Percoll gradient centrifugation." See missing word in ALL CAPS.

Response: This has now been corrected

Revisions made: Line 364, Briefly, approximately 40g of villous tissue was digested in trypsin (0.25%, Gibco) and DNase I (325 Kunits/mg tissue, Sigma) and purified over a discontinuous 10–70% Percoll gradient centrifugation.

6. Line 568 "The effects of treatment within each sex were examined post-hoc by using the two-stage linear step-up procedure of Benjamini, Krieger and Yekutieli⁴⁸ [AND] adjusted $P < 0.05$ was considered significant. See missing word in ALL CAPS.

Response: This has now been corrected

Revisions made: Line 539, The effects of treatment within each sex were examined post-hoc by using the two-stage linear step-up procedure of Benjamini, Krieger and Yekutieli⁴⁸ and adjusted $P < 0.05$ was considered significant.